# Impact of an angulated aorto-septal relationship on cardio-cerebrovascular outcomes in patients undergoing hemodialysis

Takafumi Nakayama[1,2,3]*, Junki Yamamoto[1,3], Toshikazu Ozeki[4], Shigehiro Tokoroyama[5], Yoshiko Mori[5], Mayuko Hori[5], Makoto Tsujita[5], Yuichi Shirasawa[5], Asami Takeda[5], Chika Kondo[5], Minako Murata[5], Shigeru Suzuki[5], Yuko Kinoshita[5], Michio Fukuda[5], Tsuneo Ueki[6], Noriyuki Ikehara[2], Masato Sugiura[2], Toshihiko Goto[3], Hiroya Hashimoto[7], Kazuhiro Yajima[2], Shoichi Maruyama[4], Hiroichi Koyama[8], Kunio Morozumi[5], Yoshihiro Seo[3]

1 Department of Cardiology, Masuko Memorial Hospital, Nagoya, Aichi, Japan, 2 Department of Cardiology, Nagoya City University West Medical Center, Nagoya, Aichi, Japan, 3 Department of Cardiology, Nagoya City University Graduate School of Medical Sciences, Nagoya, Aichi, Japan, 4 Department of Nephrology, Nagoya University Graduate School of Medicine, Nagoya, Aichi, Japan, 5 Department of Nephrology, Masuko Memorial Hospital, Nagoya, Aichi, Japan, 6 Department of Urology, Masuko Memorial Hospital, Nagoya, Aichi, Japan, 7 Clinical Research Management Center, Nagoya City University Hospital, Nagoya, Aichi, Japan, 8 Department of General Medicine, Masuko Memorial Hospital, Nagoya, Aichi, Japan

* tnakayama83@gmail.com

**Data Availability Statement:** All relevant data are within the paper and its Supporting Information files.

## Abstract

Aortic and valvular calcification are well-known risk factors for cardio-cerebrovascular events in patients undergoing hemodialysis. We investigated the clinical impact of an angulated aorto-septal angle as a result of aortic elongation due to aortic calcification on cardio-cerebrovascular outcomes in patients undergoing hemodialysis. We investigated 306 patients (mean age 65.4 years, 68% male) who underwent pre-scheduled routine echocardiography between April and September 2018. The angle between the anterior wall of the aorta and the ventricular septal surface (ASA) was quantified. We determined aortic and mitral valve calcification scores based on calcified cardiac changes; the aortic and mitral valve scores ranged between 0–9 and 0–6, respectively. The primary endpoint was a composite including cardio-cerebrovascular events and cardio-cerebrovascular death. The mean duration of dialysis among the patients in this analysis was 9.6 years. The primary endpoint was observed in 54 patients during the observational period (median 1095 days). Multivariable Cox proportional hazards analyses identified left ventricular ejection fraction (per 10% increase: hazard ratio [HR] 0.67; 95% confidential interval [CI] 0.53–0.84, P = 0.001), left ventricular mass index (per 10 g/m$^2$ increase: HR 1.14; 95% CI 1.05–1.24, P = 0.001), ASA (per 10 degree increase: HR 0.69; 95% CI 0.54–0.88; P = 0.003), and aortic valve calcification score (HR 1.15; 95% CI 1.04–1.26, P = 0.005) as independent determinants of the primary endpoint. Kaplan-Meier analysis showed a higher incidence of the primary endpoint in patients with ASA <119.4 degrees than those with ASA ≥119.4 degrees (Log-rank P < 0.001). An angulated aorto-septal angle is an independent risk factor for

**Funding:** The APC was funded by Japan Society for the Promotion of Science (JSPS; KAKENHI grant number 21K16191). The funders had no role in the study design, data collection and analysis, decision to publish, or preparation of the manuscript.

**Competing interests:** The authors have declared that no competing interests exist.

cardio-cerebrovascular events and cardio-cerebrovascular death in patients undergoing hemodialysis.

## 1. Introduction

Clinical management of patients undergoing hemodialysis has been recognized as important because an increasing number of patients has required hemodialysis during the last few decades [1]. Cardio-cerebrovascular events develop significantly more frequently in patients undergoing hemodialysis than those who are not [2–9], and these diseases are the main causes of death or decreased quality of life in patients requiring hemodialysis [5, 8]. Calcified valvular change and arterial calcification are well-known risk factors for cardio-cerebrovascular diseases in patients undergoing hemodialysis [10–14]. In addition, systemic calcification is one of the changes of aging and improves with the initiation of hemodialysis [15]. Aortic elongation is also a result of the aging process associated with aortic calcification [16–18]. Aortic wedging, which is considered to be accompanied by aortic elongation, could present as a narrowed angle between the left ventricular (LV) and aortic axis [19, 20]. Some recent studies demonstrated that the echocardiographic-derived angulated relationship between the ventricular septum and aorta impact poor cardio-cerebrovascular outcomes [21, 22]. Therefore, we expect that the presence of an angulated LV septum and aorta on echocardiography is a risk factor for poor cardio-cerebrovascular outcomes in patients requiring hemodialysis.

Accordingly, the purpose of the current study was to clarify the impact of sigmoid septum on cardio-cerebrovascular outcomes in patients undergoing hemodialysis.

## 2. Material and methods

### 2.1 Study population

Prescheduled routine echocardiography is performed at least once a year in patients requiring hemodialysis at Masuko Memorial Hospital. We reviewed all echocardiographic examinations performed from April to September 2018 and found 372 patients who underwent routine echocardiography. We excluded 5 patients undergoing hybrid dialysis, 16 patients due to a history of open heart surgery, and 45 patients whose echocardiographic images were not available for quantifying sigmoid septum or calcification of aortic valve and mitral valve. Finally, 306 patients were eligible for analysis in the current study (Fig 1). This study was a sub-analysis of a previous study investigating the impact of LV hypertrophy in hemodialysis patients [23]. The data were accessed for research purposes between November 6th, 2019, and September 15th, 2023, and the authors accessed information that could identify individual participants during data collection.

### 2.2 Study protocol

We retrospectively collected all clinical data and clinical outcomes from the patients' medical records, including all summaries and medical information sheets for all patients. Blood pressure and pulse rate were measured at the start of the last dialysis session before the routine echocardiographic examination. All blood examinations were performed immediately before the daily dialysis. We categorized the primary causes of dialysis into two groups: diabetes mellitus and non-diabetes mellitus. A history of cerebrovascular disease was defined as previous cerebral infarction, cerebral hemorrhage, or subarachnoid hemorrhage.

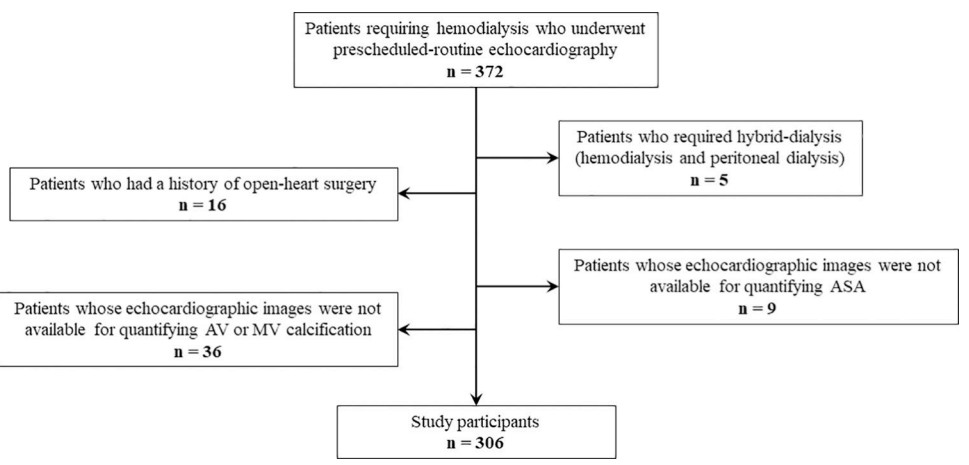

**Fig 1. Flowchart of patient enrollment.**

Hypertension, hyperlipidemia, and diabetes mellitus were included as cardiovascular risk factors. Hypertension was defined as having a history of hypertension or being under anti-hypertensive treatment. A history of hypertension is diagnosed as an office blood pressure $\geq$ 140/90 mmHg or home blood pressure $\geq$ 135/85 mmHg [24]. Hyperlipidemia was defined as having a history of hyperlipidemia, taking medication for hyperlipidemia, or a serum low-density lipoprotein level $\geq$ 140 mg/dL at baseline. Diabetes mellitus was defined as having a history of diabetes mellitus or undergoing treatment with glucose-lowering therapy and was diagnosed by meeting the following criteria at least two times, including the blood glucose criterion at least once, the year prior to echocardiography: hemoglobin A1c $\geq$ 6.5% and blood glucose $\geq$ 200 mg/dL [25].

The primary endpoint was a composite of admission and death due to cardio-cerebrovascular events, including heart failure, myocardial ischemic events, or cerebrovascular events. A myocardial ischemic event was defined as an admission due to ischemic heart disease that required revascularization. A cerebrovascular event was defined as cerebral infarction, cerebral hemorrhage, or subarachnoid hemorrhage. Unexplained sudden death and any death accompanied by ventricular arrhythmia were included in cardio-cerebrovascular death. Two cardiologists (TN and JY) and a nephrologist (TO) carefully assessed the cardio-cerebrovascular events from medical records, including all summaries, medical information sheets, and reference letters. If the authors needed to clarify the details, they contacted a neurologist or the attending hemodialysis physician at Masuko Memorial Hospital to determine an accurate endpoint.

The Institutional Ethical Review Board of Masuko Memorial Hospital approved this study (no. MR1-18) and waived informed consent. The information on this study is available from the Masuko Memorial Hospital website, and all patients were given the opportunity to withdraw from the study. This retrospective study was conducted according to the principles of the Declaration of Helsinki.

## 2.3 Echocardiographic measurements

We retrospectively obtained the echocardiographic parameters from the prescheduled routine echocardiography reports. All echocardiography was performed and assessed under the recommendations of the American Society of Echocardiography [26]. LV ejection fraction (LVEF) was calculated using the Teichholz method. LV wall thickness was the mean value of

the intraventricular septal diameter (IVSd) and posterior wall diameter (PWd). LV mass was calculated using LV diastolic diameter (LVDd) and the following formula [26]:

$$\text{LV mass} = 1.04 \times \left( \left( \text{LVDd} + \text{IVSd} + \text{PWd} \right)^3 - \text{LVDd}^3 \right) \times 0.8 + 0.6) \times 0.001.$$

The severity of stenotic valvular disease was evaluated at the aortic and mitral valves according to the recommendations of the American Society of Echocardiography [26] independently from the valve calcification scoring below. The grade of regurgitative valvular disease was assessed semi-quantitatively by sonographers.

We determined the date-interval between the day of echocardiographic examination and the last dialysis before echocardiography as follows: 0, echocardiography was examined after dialysis on the same day; 1–3, echocardiography was examined 1–3 days after the last dialysis.

## 2.4 Echocardiographic quantification of the angle between the aorta and left ventricle

We measured the angle between the anterior wall of the aorta and the ventricular septal surface (aorto-septal angle; ASA) to assess the degree of angulation between the aorta and left ventricle (Fig 2A). A lower ASA means greater morphological changes. To evaluate intra- and inter-rater reliabilities, the ASA was remeasured by the first rater (T.N.) and the second rater (J.Y.) in 50 patients who underwent echocardiography between August and September 2018.

## 2.5 Echocardiographic scoring of aortic and mitral valve calcification

We scored aortic valve leaflet calcification as follows: 0, no calcification on the cusp and leaflet; 1, calcification only on the tip of the leaflet; 2, calcification on the tip and mid-portion or base of the leaflet with a free opening; 3, calcification on the leaflet with a limited opening (Fig 2B). We also scored mitral valve leaflet calcification as follows: 0, no calcification on the leaflet and annulus; 1, calcification only on the annulus; 2, calcification on the annulus and mid-portion of the leaflet with a free opening; 3, calcification on the whole leaflet with a limited opening (Fig 2C). We defined the aortic valve calcification score (AVCS) as the sum of the three aortic leaflets' scores and the mitral valve calcification score (MVCS) as the sum of two mitral leaflets' scores.

## 2.6 Statistical analysis

Continuous variables are presented as the mean ± standard deviation. We compared continuous variables between groups using the unpaired Student's t-test when normally distributed, and Mann-Whitney test when not normally distributed. Comparisons of categorical variables were assessed with the Pearson's chi-squared test. We used Cox proportional hazards analysis to identify the risk factors for the primary endpoint. The initial time-point for the survival analysis was the date of baseline echocardiography (prescheduled routine echocardiography). Cox proportional hazards analyses were performed using ASA, AVCS, MVCS, and other echocardiographic and clinical variables, including parameters that are considered risk factors for the primary endpoint [2, 23, 27–33]. Regarding the incidence of the primary endpoint, up to five variables that achieved P-values < 0.05 in the univariable analyses were available in the multivariable Cox proportional hazards analyses. First, we assessed echocardiographic parameters in multivariable analyses to determine the independent risk factors for the primary endpoint, then we performed additional analyses including the other clinical variables that had P < 0.20 as supplemental information. When the clinical meaning was almost the same, or a correlation coefficient > 0.70 or < -0.70 was found, one of the values was excluded from the

A

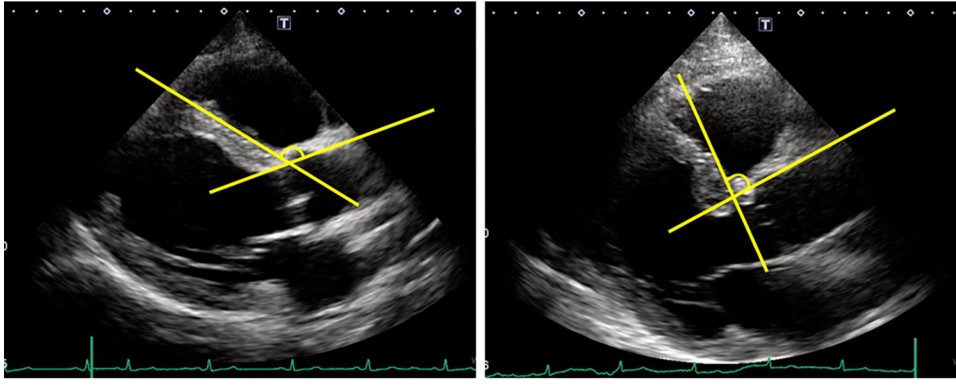

ASA: 127 degree          ASA: 86 degree

B

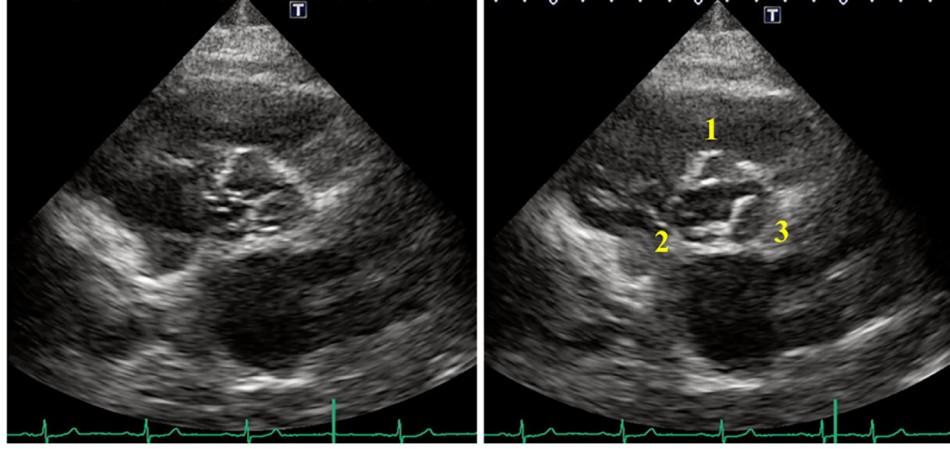

C

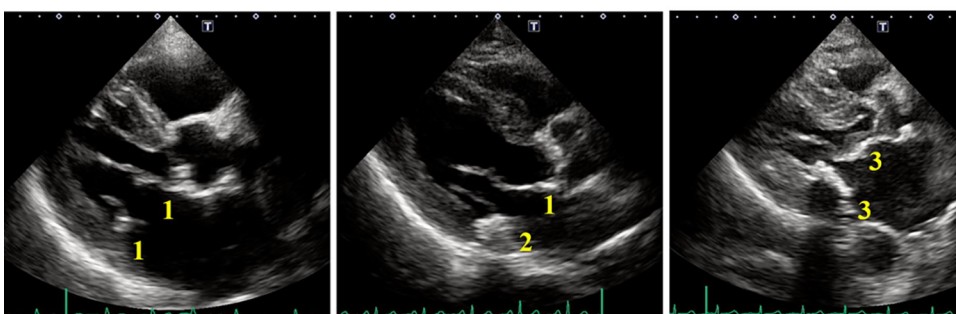

**Fig 2.** Representative images for quantifying the angle between the anterior wall of the aorta and the ventricular septal surface (A), aortic valve calcification score (B), and mitral valve calcification score (C). (B); The right coronary cusp has calcification only on the tip (score = 1), the non-coronary cusp has calcification on the tip to mid of leaflet with a free opening (score = 2), and the left coronary cusp has calcification and limited opening (score = 3), accordingly AVCS is 6. (C) left panel; both anterior and posterior mitral valve leaflets have calcification only on annulus (MVCS = 1+1 = 2), (C) middle panel; anterior leaflet has calcification on annulus and posterior leaflet has calcification on annulus to leaflet with a free opening (MVCS = 1+2 = 3), (C) right panel; both anterior and posterior leaflet have calcification on annulus and leaflet with limited opening (MVCS = 3+3 = 6).

multivariable Cox proportional hazards analyses. Spearman's test was used to analyze the relationships between the echocardiographic parameters and the dialytic date-interval. Kaplan-Meier curves were used to compare the event-free survival rate between groups with the log-rank test. The cut-off value was determined by Youden's index, which was derived with the time-dependent receiver operating characteristics (ROC) curve at 36 months. In this retrospective study, patient prognosis was evaluated up to 36 months. To clarify the factors associated with the aortic wedging, we analyzed the univariable and multivariable logistic regression analyses for an ASA below the cut-off value. We calculated intraclass correlation coefficients (ICC) to evaluate intra- and inter-rater reliabilities, and an ICC $\geq$ 0.80 was assessed as the preferred level of reliability [34].

Two-sided P-values < 0.05 were considered significant. All analyses were performed using SPSS ver.26 (IBM Corp., Armonk, NY, USA) and SAS ver. 9.4 (SAS Institute, Cary, NC, USA).

## 3. Results

### 3.1 Baseline characteristics

The baseline characteristics are summarized in Table 1. The patient group that underwent the primary endpoint had higher age, smaller ASA, and greater AVCS and MVCS. The ICCs for the intra- and inter-rater reliabilities for measuring ASA were 0.88 and 0.81, respectively.

### 3.2 Risk factors for the primary endpoint

The primary endpoint was observed in 54 patients during the observational period, which was a median 1095 days (interquartile range, 908–1095 days). Univariable Cox proportional hazards analyses revealed 10 echocardiographic parameters with p < 0.05: LVEF, LV systolic diameter, LV wall thickness, LV mass index, left atrial diameter, early diastolic transmitral inflow velocity, ratio of the early diastolic transmitral flow velocity to mitral annular velocity, AVCS, and MVCS. Multivariable Cox proportional hazards analyses identified LVEF (per 10% increase: hazard ratio [HR] 0.67; 95% confidential interval [CI] 0.53–0.84; P = 0.001), LV mass index (per 10 g/m$^2$ increase: HR 1.14; 95% CI 1.05–1.24; P = 0.001), ASA (per 10 degree increase; HR 0.69; 95% CI 0.54–0.88; P = 0.003), and AVCS (HR 1.15; 95% CI 1.04–1.26; P = 0.005) as independent determinants of the primary endpoint (Table 2).

The univariable Cox proportional hazards analyses also revealed six clinical variables (age, history of diabetes mellitus, history of ischemic heart disease or cerebrovascular disease, serum albumin level, serum sodium level, oral administration of anti-platelet or anti-coagulation therapy) that were significantly associated with the primary endpoint. The main results did not change when each of the six clinical parameters were also included in the multivariable Cox proportional hazards analyses (S1 Table).

The time-dependent ROC curve-derived cut-off value of ASA was 119.4 degrees (S1 Fig). Kaplan-Meier analysis demonstrated that the lower ASA group had a significantly higher incidence of the primary endpoint compared to the higher ASA group (Log-rank P < 0.001, Fig 3).

### 3.3 Factors associated with an angulated aorto-septal angle

The univariable logistic regression analyses revealed seven variables (age, pulse rate, serum albumin level, serum sodium level, anti-platelet or anti-coagulation therapy, AVCS, and MVCS) that were significantly associated with ASA < 119.4. Only age (odds ratio, 1.41; 95% CI 1.11–1.80; P = 0.005) was found to be an independent determinant of ASA < 119.4 in the multivariable logistic regression analysis (Table 3).

**Table 1. Baseline characteristics of the study population.**

| Characteristic | All patients | Event free | Event occurrence | P value |
|---|---|---|---|---|
| | n = 306 | n = 252 | n = 54 | |
| Echocardiography | | | | |
| LVEF, % | 65.6 ± 10.0 | 66.8 ± 9.1 | 60.1 ± 12.1 | < 0.001 |
| LV diastolic diameter, mm | 47.6 ± 6.2 | 47.3 ± 6.1 | 49.2 ± 6.3 | 0.036 |
| LV systolic diameter, mm | 30.3 ± 6.2 | 29.7 ± 5.9 | 33.3 ± 6.8 | < 0.001 |
| LV wall thickness, mm | 10.1 ± 1.8 | 10.0 ± 1.8 | 10.9 ± 1.8 | 0.001 |
| LV mass index, g/m$^2$ | 109.2 ± 32.7 | 105.5 ± 31.7 | 126.1 ± 32.3 | < 0.001 |
| Left atrial diameter, mm | 36.2 ± 6.2 | 35.8 ± 6.1 | 38.2 ± 6.4 | 0.009 |
| E wave, cm/s | 76.0 ± 23.6 | 74.9 ± 21.5 | 81.3 ± 31.2 | 0.073 |
| Deceleration time, ms | 228.7 ± 64.5 | 229.5 ± 60.7 | 224.9 ± 80.6 | 0.64 |
| E/E' (n = 258) | 11.5 ± 4.8 | 10.9 ± 4.1 | 14.2 ± 6.6 | < 0.001 |
| Valvular disease ≥ moderate | 36 (12%) | 23 (9%) | 13 (24%) | 0.002 |
| Aortic valve regurgitation | 7 (2%) | 4 (2%) | 3 (6%) | 0.089 |
| Mitral valve regurgitation | 17 (6%) | 12 (5%) | 5 (9%) | 0.19 |
| Tricuspid valve regurgitation | 13 (4%) | 7 (3%) | 6 (11%) | 0.006 |
| Pulmonary valve regurgitation | 2 (1%) | 2 (1%) | 0 (0%) | 0.50 |
| Aortic valve stenosis | 4 (1%) | 2 (1%) | 2 (4%) | 0.088 |
| Mitral valve stenosis | 4 (1%) | 2 (1%) | 2 (4%) | 0.088 |
| ASA | 117.9 ± 11.4 | 119.0 ± 11.0 | 112.9 ± 12.1 | < 0.001 |
| AV calcification score | | | | 0.005 |
| 0–2 | 198 | 171 (68%) | 27 (50%) | |
| 3–5 | 68 | 55 (22%) | 13 (24%) | |
| 6–9 | 40 | 26 (10%) | 14 (26%) | |
| MV calcification score | | | | < 0.001 |
| 0 | 202 | 180 (71%) | 22 (41%) | |
| 1 | 69 | 48 (19%) | 21 (39%) | |
| 2–6 | 35 | 24 (10%) | 11 (20%) | |
| Basic data | | | | |
| Age, y | 65.4 ± 12.7 | 64.3 ± 12.7 | 70.6 ± 11.1 | 0.001 |
| Male | 209 (68%) | 169 (68%) | 40 (71%) | 0.58 |
| BMI, kg/m$^2$ | 21.8 ± 4.4 | 21.7 ± 4.1 | 22.8 ± 5.4 | 0.094 |
| Blood pressure, mmHg | 147.2 ± 22.9 | 146.2 ± 22.6 | 151.8 ± 23.9 | 0.10 |
| Pulse rate, bpm | 72.8 ± 11.8 | 72.5 ± 11.8 | 74.2 ± 11.9 | 0.34 |
| Dialysis duration, y | 9.6 ± 8.9 | 9.7 ± 9.1 | 8.9 ± 8.4 | 0.54 |
| Primary disease of dialysis | | | | |
| Diabetes mellitus | 95 (31%) | 74 (29%) | 21 (39%) | 0.17 |
| Cardiovascular risk factors | | | | |
| Hypertension | 225 (74%) | 185 (73%) | 40 (74%) | 0.92 |
| Dyslipidemia | 103 (34%) | 85 (34%) | 18 (33%) | 0.96 |
| Diabetes mellitus | 118 (39%) | 90 (36%) | 28 (52%) | 0.027 |
| History of ischemic heart disease | 41 (13%) | 27 (11%) | 14 (26%) | 0.003 |
| History of cerebrovascular disease | 29 (9%) | 19 (8%) | 10 (19%) | 0.012 |
| Laboratory measurements | | | | |
| Hemoglobin, g/dL | 11.2 ± 1.1 | 11.2 ± 1.2 | 11.2 ± 1.0 | 0.94 |
| Platelets, *10$^4$/μg | 19.0 ± 6.1 | 19.1 ± 6.3 | 18.1 ± 5.4 | 0.24 |
| Albumin, g/dL | 3.6 ± 0.4 | 3.6 ± 0.4 | 3.5 ± 0.3 | 0.067 |
| Total bilirubin, mg/dL | 0.3 ± 0.1 | 0.3 ± 0.1 | 0.3 ± 0.1 | 0.11 |

(*Continued*)

**Table 1.** (Continued)

| Characteristic | All patients | Event free | Event occurrence | P value |
|---|---|---|---|---|
| | n = 306 | n = 252 | n = 54 | |
| LDH, IU/L | 181.9 ± 35.0 | 182.2 ± 35.2 | 180.5 ± 34.7 | 0.75 |
| BUN, mg/dL | 57.7 ± 14.3 | 58.0 ± 13.8 | 56.6 ± 16.7 | 0.53 |
| Serum sodium, mEq/L | 138.7 ± 3.1 | 138.9 ± 3.1 | 138.0 ± 3.1 | 0.041 |
| Serum calcium, mg/dL | 8.6 ± 0.7 | 8.6 ± 0.7 | 8.6 ± 0.5 | 0.87 |
| Serum phosphorus, mmol/L | 5.4 ± 5.3 | 5.4 ± 5.8 | 5.1 ± 1.3 | 0.67 |
| Serum Int-PTH pg/mL | 136.5 (86.8–208.0) | 137.0 (84.8–208.0) | 134.0 (87.5–201.5) | 0.84 |
| Medication | | | | |
| B-blocker | 102 (33%) | 81 (32%) | 21 (39%) | 0.34 |
| ACEI/ARB | 135 (44%) | 110 (44%) | 25 (46%) | 0.72 |
| Ca-blocker | 165 (54%) | 142 (56%) | 23 (43%) | 0.066 |
| Statin | 77 (25%) | 63 (25%) | 14 (26%) | 0.89 |
| Anti-platelet or anti-coagulation | 102 (33%) | 72 (29%) | 30 (56%) | < 0.001 |

LV, left ventricular; EF, ejection fraction; E wave, the early diastolic transmitral flow velocity; E/E', ratio of the early diastolic transmitral flow velocity to mitral annular velocity; ASA, aorto-septal angle, the angle between the anterior wall of the aorta and the ventricular septal surface; AV, aortic valvular; MV, mitral valvular; BMI, body mass index; AST, aspartate aminotransferase; ALT, alanine aminotransferase; LDH, lactate dehydrogenase; BUN, blood urea nitrogen; PTH, parathormone; ACEI, angiotensin converting enzyme inhibitor; ARB, angiotensin 2 receptor blocker.

## 4. Discussion

We demonstrated that the presence of a narrowed ASA and greater calcification of the aortic valve are independent risk factors for cardio-cerebrovascular events in patients undergoing hemodialysis. The prognostic value of a narrowed ASA in hemodialysis patients is an entirely novel finding, though it is understandable because of its substrates being associated with aging and changes in calcification. Additional multivariable Cox proportional hazards analyses showed that the results did not change when we included the other clinical variables significant in the univariable analysis (S1 Table).

### 4.1 Angulated aorto-septal angle

The clinical evidence for an angulated ASA is still small but has recently grown. An association with clinically adverse events and diastolic dysfunction has been reported in non-hemodialysis patients [21, 22, 35]. Aortic wedging that correlates with an angulated LV-aorta angle is strongly associated with age [20]. Our investigations also showed that only age is an independent determinant of ASA ≤ 119.4 degrees (Table 3). However, as both the presence of an angulated ASA and higher age were independent risk factors for cardio-cerebrovascular events, an angulated ASA must have unexplained factors other than age causing adverse events. For example, a narrowed ASA is associated with LV diastolic dysfunction [35], increased central blood pressure, and increased aortic pressure wave reflection [36], which can affect the incidence of cardio-cerebrovascular events. Furthermore, continuous exposure to high blood pressure, higher aortic artery calcification volume, and hemodynamic disadvantage due to non-straight LV outlet are possibly associated with cumulative myocardial and vascular injury. As AVCS and MVCS were significantly associated with a narrowed ASA in the univariable logistic regression analysis, valvular calcification may morphologically contribute to the angulated ASA during the aging process. In the current study, no patients had a significant LV outlet tract obstruction (pressure gradient ≥ 50 mmHg [37]) with systolic anterior-mitral valve leaflet motion during routine echocardiographic examination, and LV outlet tract obstruction

**Table 2. Cox proportional hazards analyses.**

| Characteristic | Univariable analyses | | | Multivariable analyses | | |
|---|---|---|---|---|---|---|
| | HR | 95% CI | P value | HR | 95% CI | P value |
| Echocardiography | | | | | | |
| LVEF, per 10-% increase | 0.62 | 0.48–0.75 | < 0.001 | 0.67 | 0.53–0.84 | 0.001 |
| LV diastolic diameter, mm | 1.04 | 1.00–1.09 | 0.052 | | | |
| LV systolic diameter, mm | 1.08 | 1.03–1.12 | < 0.001 | - | - | - |
| LV wall thickness, mm | 1.24 | 1.10–1.41 | 0.001 | - | - | - |
| LV mass index, per 10-g/m² increase | 1.18 | 1.10–1.28 | < 0.001 | 1.14 | 1.05–1.24 | 0.001 |
| Left atrial diameter, mm | 1.07 | 1.02–1.11 | 0.003 | - | - | - |
| E wave, per 10-cm/s increase | 1.12 | 1.01–1.25 | 0.036 | - | - | - |
| Deceleration time, per 10-ms increase | 0.99 | 0.95–1.03 | 0.61 | | | |
| E/E' (n = 258) | 1.11 | 1.06–1.16 | < 0.001 | - | - | - |
| ASA, per 10-degree increase | 0.64 | 0.51–0.81 | < 0.001 | 0.69 | 0.54–0.88 | 0.003 |
| AV calcification score | 1.20 | 1.10–1.32 | < 0.001 | 1.15 | 1.04–1.26 | 0.005 |
| MV calcification score | 1.48 | 1.20–1.83 | < 0.001 | 1.25 | 0.98–1.61 | 0.072 |
| Basic data | | | | | | |
| Age, per 10-year increase | 1.59 | 1.25–2.01 | < 0.001 | - | - | - |
| Male, vs female | 1.19 | 0.66–2.17 | 0.56 | | | |
| BMI, kg/m² | 1.04 | 0.99–1.10 | 0.16 | | | |
| Blood pressure, per 10-mmHg increase | 1.10 | 0.98–1.24 | 1.10 | | | |
| Pulse rate, per 10-bpm increase | 1.12 | 0.90–1.40 | 0.33 | | | |
| Dialysis duration, y | 0.99 | 0.96–1.02 | 0.54 | | | |
| Primary disease of dialysis | | | | | | |
| Diabetes mellitus, vs. non-diabetes mellitus | 1.52 | 0.88–2.63 | 0.13 | | | |
| Cardiovascular risk factors | | | | | | |
| Hypertension | 1.00 | 0.55–1.84 | 1.00 | | | |
| Dyslipidemia | 0.95 | 0.54–1.68 | 0.87 | | | |
| Diabetes mellitus | 1.81 | 1.06–3.01 | 0.030 | - | - | - |
| History of ischemic heart disease or cerebrovascular disease | 2.42 | 1.38–4.23 | 0.002 | - | - | - |
| Laboratory measurements | | | | | | |
| Hemoglobin, g/dL | 0.98 | 0.77–1.24 | 0.84 | | | |
| Platelets, *10⁴/μg | 0.97 | 0.93–1.02 | 0.25 | | | |
| Albumin, g/dL | 0.44 | 0.23–0.85 | 0.015 | - | - | - |
| Total bilirubin, per 0.1-mg/dL increase | 1.17 | 0.98–1.41 | 0.085 | | | |
| Serum sodium, mEq/L | 0.92 | 0.85–0.99 | 0.023 | - | - | - |
| Serum potassium, mEq/L | 0.70 | 0.47–1.04 | 0.074 | | | |
| Serum calcium, mg/dL | 1.03 | 0.70–1.50 | 0.90 | | | |
| Serum phosphorus, mmol/L | 0.97 | 0.86–1.11 | 0.69 | | | |
| Serum Int-PTH, per 10-pg/mL increase | 0.99 | 0.97–1.02 | 0.57 | | | |
| Medication | | | | | | |
| B-blocker | 1.28 | 0.74–2.22 | 0.37 | | | |
| ACEI/ARB | 1.05 | 0.62–1.80 | 0.85 | | | |
| Ca-blocker | 0.61 | 0.36–1.05 | 0.074 | | | |
| Anti-platelet or anti-coagulation | 2.87 | 1.68–4.92 | < 0.001 | - | - | - |

FLV, left ventricular; EF, ejection fraction; E wave, the early diastolic transmitral flow velocity; E/E', ratio of the early diastolic transmitral flow velocity to mitral annular velocity; ASA, aorto-septal angle, the angle between the anterior wall of the aorta and the ventricular septal surface; AV, aortic valvular; MV mitral valvular; BMI, body mass index; PTH, parathormone; ACEI, angiotensin converting enzyme inhibitor; ARB, angiotensin 2 receptor blocker.

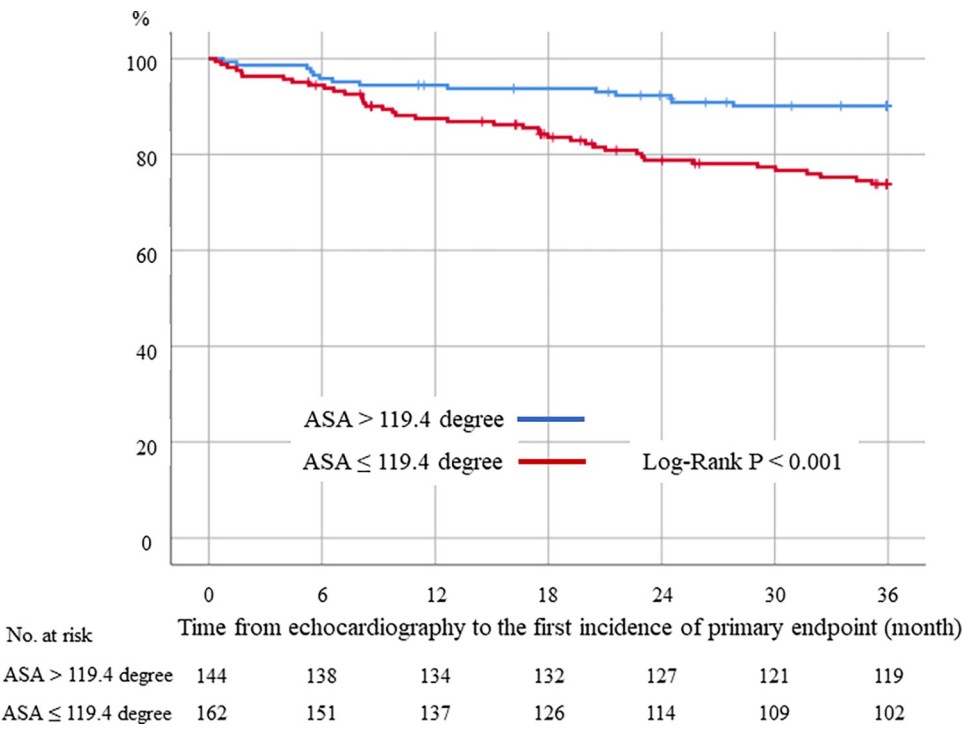

**Fig 3. Kaplan-Meier curves for the primary endpoint in the two groups according to the aorto-septal angle (ASA) cut-off value of 119.4 degrees.** Patients with ASA ≥ 119.4 degrees demonstrated significantly lower incidence of the primary endpoint than those with ASA < 119.4 degrees.

was not associated with our results. Thus, the presence of an angulated ASA must be a surrogate marker of negative factors that accumulate and are affected not only by aging, but also dialysis-associated changes in calcification and tissue injury.

On the other hand, sigmoid septum is a well-known morphological change similar to, but different from, an angulated ASA. Initially, sigmoid septum was defined as the base of the ventricular septum protruding toward the LV cavity [38]. Sigmoid septum, which is sometimes referred to as subaortic ventricular septal bulge or discrete upper septal hypertrophy, is associated with age [39–41]. In addition, atherosclerosis, thickening of the aortic or mitral valve, and hypertension are considered to contribute to the presence of sigmoid septum [42–44]. Otherwise, sigmoid septum has no impact on the cardiovascular and mortality risk in patients with heart failure and on exercise tolerance in healthy populations [45, 46]. Interestingly, an angulated relationship between the ventricular septum and ascending aorta did not correlate with the extent of septal bulge in prior research using computed tomography [47], suggesting that the underlying substrates and pathological roles are different between an angulated ASA and sigmoid septum. For example, aortic elongation is not always accompanied by atherosclerosis [17, 18]. Referring to the evidence above and the results of the current study, an angulated ASA is likely to affect the clinical prognosis compared with sigmoid septum. Further research investigating the difference in contributing factors and influence between an angulated ASA and sigmoid septum in various groups of patients is of interest. In a previous study, we used ASA as a representative parameter of sigmoid septum [21]. However, as we updated our knowledge with the latest research, we came to appreciate these two different morphological concepts; it is correct to use ASA to indicate aortic wedging due to aortic elongation. Aortic wedging and sigmoid septum look similar but, as mentioned above, these two concepts must be distinguished regarding different substrates and likely different clinical impacts.

**Table 3. Logistic regression analysis of lower ASA (<119.4 degrees).**

| Characteristic | Univariable analyses | | | Multivariable analyses | | |
|---|---|---|---|---|---|---|
| | OR | 95% CI | P value | OR | 95% CI | P value |
| Basic data | | | | | | |
| Age, per 10-year increase | 1.68 | 1.37–2.05 | < 0.001 | 1.42 | 1.11–1.81 | 0.006 |
| Male, vs female | 0.96 | 0.59–1.56 | 0.87 | | | |
| Height, cm | 0.98 | 0.96–1.00 | 0.076 | | | |
| Weight, kg | 1.00 | 0.98–1.01 | 0.49 | | | |
| BMI, kg/m$^2$ | 1.01 | 0.96–1.06 | 0.82 | | | |
| Blood pressure, per 10-mmHg increase | 1.01 | 0.92–1.12 | 0.83 | | | |
| Pulse rate, per 10-bpm increase | 0.82 | 0.67–0.99 | 0.042 | 0.97 | 0.77–1.21 | 0.76 |
| Dialysis duration, y | 0.99 | 0.97–1.02 | 0.49 | | | |
| Primary disease of dialysis | | | | | | |
| Diabetes mellitus, vs. non-diabetes mellitus | 1.34 | 0.82–2.18 | 0.25 | | | |
| Cardiovascular risk factors | | | | | | |
| Hypertension | 1.21 | 0.73–2.02 | 0.46 | | | |
| Dyslipidemia | 0.97 | 0.60–1.56 | 0.90 | | | |
| Diabetes mellitus | 1.22 | 0.77–1.93 | 0.41 | | | |
| History of cardiovascular disease | 1.86 | 0.93–3.70 | 0.078 | | | |
| History of cerebrovascular disease | 1.29 | 0.59–2.80 | 0.52 | | | |
| History of atrial fibrillation | 1.12 | 0.43–2.92 | 0.82 | | | |
| Laboratory measurements | | | | | | |
| Hemoglobin, g/dL | 0.91 | 0.73–1.10 | 0.33 | | | |
| Platelets, *10$^4$/μg | 1.00 | 0.97–1.04 | 0.82 | | | |
| Albumin, g/dL | 0.36 | 0.19–0.71 | 0.003 | 0.74 | 0.35–1.58 | 0.44 |
| Total bilirubin, mg/dL | 1.01 | 0.17–5.91 | 0.99 | | | |
| Serum sodium, mEq/L | 0.91 | 0.84–0.98 | 0.015 | 0.93 | 0.85–1.01 | 0.090 |
| Serum calcium, mg/dL | 0.76 | 0.53–1.09 | 0.14 | | | |
| Serum phosphorus, mmol/L | 0.97 | 0.88–1.06 | 0.45 | | | |
| Medication | | | | | | |
| B-blocker | 0.70 | 0.44–1.13 | 0.15 | | | |
| ACEI/ARB | 1.03 | 0.65–1.62 | 0.90 | | | |
| Anti-platelet or anti-coagulation | 1.82 | 1.12–2.96 | 0.016 | 1.34 | 0.82–2.36 | 0.23 |
| Echocardiography | | | | | | |
| LVEF, per 10-% increase | 0.92 | 0.73–1.15 | 0.44 | | | |
| LV diastolic diameter, mm | 0.99 | 0.96–1.03 | 0.61 | | | |
| LV systolic diameter, mm | 1.00 | 0.96–1.04 | 1.00 | | | |
| LV wall thickness, mm | 1.05 | 0.93–1.19 | 0.44 | | | |
| LV mass index, per 10-g/m$^2$ increase | 1.02 | 0.95–1.09 | 0.63 | | | |
| Left atrial diameter, mm | 1.03 | 1.00–1.07 | 0.083 | | | |
| E wave, per 10-cm/s increase | 0.96 | 0.87–1.05 | 0.36 | | | |
| Deceleration time, per 10-ms increase | 1.04 | 1.00–1.08 | 0.049 | 1.02 | 0.98–1.06 | 0.32 |
| E/E' (n = 258) | 1.03 | 0.98–1.09 | 0.28 | | | |
| Valvular disease ≥ moderate | 1.91 | 0.92–3.98 | 0.083 | | | |
| AV calcification score | 1.19 | 1.08–1.32 | < 0.001 | 1.06 | 0.94–1.18 | 0.34 |
| MV calcification score | 1.53 | 1.12–2.09 | 0.007 | 1.18 | 0.85–1.65 | 0.33 |

BMI, body mass index; ACEI, angiotensin converting enzyme inhibitor; ARB, angiotensin 2 receptor blocker; LV, left ventricular; EF, ejection fraction; E wave, the early diastolic transmitral flow velocity; E/E', ratio of the early diastolic transmitral flow velocity to mitral annular velocity; AV, aortic valvular; MV mitral valvular.

Sigmoid septum is associated with the presence of Q waves in V1 and V2 leads [48]. As findings of ischemic heart disease are clinically essential for patients undergoing dialysis because of their high prevalence of ischemic heart disease [49, 50], we investigated the association between ASA and the presence of Q wave in V1-2. As a result, the average ASAs were 114.0 ± 13.1 degrees and 118.2 ± 11.3 degrees in patients with Q wave in both V1-2 leads (n = 20) and patients without Q wave (n = 286), and the difference was not significant (P = 0.12). The above results may be affected by the small number of participants; ASA was not a representative measurement of sigmoid septum. The presence of a non-ischemic Q wave in dialysis patients is essential to clinically differentiate in this cohort; thus, further studies in a larger group of patients undergoing dialysis are needed to investigate the relationship between the representative measurements of sigmoid septum and Q wave in V1-2.

It seemed reproducible to measure ASA regarding the preferred ICCs for intra- and inter-reliabilities of 0.88 and 0.81, respectively. The previous research also supported a high reproducibility in measuring ASA [21]. We checked the association between echocardiographic parameters and dialytic date-interval (S2 Table). Significant correlations were observed in volume parameters (LV diastolic and systolic diameters and left atrial diameter) and ASA, but not in morphological parameters (LVEF, LV wall thickness, and LV mass index; S2A Table). ASA and dialytic interval had a positive correlation coefficient, suggesting that the hypovolemic condition possibly narrowed the ASA. After including dialytic date-interval in the multivariable Cox proportional hazards analysis, the main results were the same (S2B Table).

## 4.2 Aortic and mitral valve calcification

Previous studies demonstrated that hemodialysis patients with valvular calcification have worse clinical outcomes than those without valvular calcification [10–12]. The associated factors for calcification are different between AV and MV [51]. AV calcification is affected by the same risk factors for systemic vascular calcification [52], such as age and higher serum calcium levels. A higher incidence of cardio-cerebrovascular events is considered reasonable in patients with higher AVCS given these backgrounds. We must differentiate aortic valve calcification from aortic valve stenosis (AS). In the current study, 4 patients had moderate or greater AS (S3A Table). When adding AS ≥ moderate to the multivariable analysis, the main result was the same (S3B Table). Thus, we can guess that the worse cardio-cerebrovascular outcomes in higher AVCS patients were independent of AS-associated hemodynamics.

On the other hand, MV calcification was not associated with the factors above and was not an independent determinant of cardio-cerebrovascular events in the current study. Ikee et al. reported that MV calcification is associated with serum β2 microglobulin levels, which may reflect various harmful conditions for the cardiovascular system [51] and inflammation in patients requiring hemodialysis [53, 54]. Moreover, β2 microglobulin is considered to indicate hemodialysis clearance of middle molecules. As AV calcification was more frequently observed than MV calcification in patients with chronic renal failure not requiring hemodialysis [55], MV calcification probably develops after initiating hemodialysis [51]. In our country, the strict and superior quality of hemodialysis management evidenced by a previous study [56] may result in lower serum β2 microglobulin and controlled MV calcification.

## 4.3 Study limitations

The current study has some limitations. The investigation was retrospective at a single center. The sample size and incidence of the primary endpoint were not enough, resulting in the availability of a limited number of variables to enter into the multivariable analyses. Data on serum BNP levels were missing. the date-interval between the echocardiographic examination and

nearest hemodialysis varied. As the LVOT pressure gradient was quantified in only necessary patients, we could not perform a dynamic evaluation of the extent of LVOT narrowing. Furthermore, because exercise or dobutamine stress echocardiography were not performed at Mesuko Memorial Hospital, the relationship between the ASA and subclinical LVOT obstruction was not assessed.

## 5. Conclusion

The presence of an echocardiography-derived angulated ASA and greater area of calcification on the aortic valve were independent determinants of cardio-cerebrovascular events in patients requiring hemodialysis.

## Supporting information

**S1 Fig. Receiver operating characteristics curve of the aorto-septal angle for the primary endpoint.** The optimal cut-off value was 119.4 degrees as determined by Youden's index. The sensitivity and specificity of the cut-off value were 0.48 and 0.73, respectively.
(TIF)

**S1 Table. A-I.** Additional Cox proportional hazards analyses.
(DOCX)

**S2 Table. A.** Correlations between dialytic date-interval and echocardiographic parameters. **B.** Additional Cox proportional hazards analyses.
(DOCX)

**S3 Table. A.** Aortic valve and mitral valve calcification scores for patients with moderate or greater aortic valve stenosis. **B.** Additional Cox proportional hazards analyses.
(DOCX)

**S1 Data.**
(XLSX)

## Acknowledgments

We want to acknowledge the patients who participated in this study. We also thank Emi Hanai and all of the clinical technologists, sonographers, and doctors who were associated with the patients on dialysis in Masuko Memorial Hospital.

## Author Contributions

**Conceptualization:** Takafumi Nakayama.

**Data curation:** Junki Yamamoto, Toshikazu Ozeki, Shigehiro Tokoroyama, Yoshiko Mori, Mayuko Hori, Makoto Tsujita, Yuichi Shirasawa, Asami Takeda, Chika Kondo, Minako Murata, Shigeru Suzuki, Yuko Kinoshita, Michio Fukuda, Tsuneo Ueki, Noriyuki Ikehara, Masato Sugiura, Hiroichi Koyama.

**Formal analysis:** Takafumi Nakayama, Hiroya Hashimoto.

**Funding acquisition:** Takafumi Nakayama.

**Investigation:** Takafumi Nakayama, Hiroya Hashimoto.

**Methodology:** Takafumi Nakayama.

**Supervision:** Noriyuki Ikehara, Masato Sugiura, Toshihiko Goto, Hiroya Hashimoto, Kazu-
hiro Yajima, Shoichi Maruyama, Hiroichi Koyama, Kunio Morozumi, Yoshihiro Seo.

**Writing – original draft:** Takafumi Nakayama.

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
