## [Decision Letter · Decision Letter 0]

22 Nov 2023

PONE-D-23-33216Impact of Sigmoid Septum on Cardio-cerebrovascular Outcomes in Patients Undergoing HemodialysisPLOS ONE

Dear Dr. Nakayama,

Thank you for submitting your manuscript to PLOS ONE. After careful consideration, we feel that it has merit but does not fully meet PLOS ONE’s publication criteria as it currently stands. Therefore, we invite you to submit a revised version of the manuscript that addresses the points raised during the review process.

We look forward to receiving your revised manuscript.

Kind regards,

Satoshi Higuchi

Academic Editor

PLOS ONE

Journal Requirements:

- https://doi.org/10.3390/biomedicines11020592

In your revision ensure you cite all your sources (including your own works), and quote or rephrase any duplicated text outside the methods section. Further consideration is dependent on these concerns being addressed.

"The APC was funded by JSPS KAKENHI grant Number 21K16191."

4. Please expand the acronym “JSPS” (as indicated in your financial disclosure) so that it states the name of your funders in full.

Additional Editor Comments: Thank you for your interesting work. The handling editor would like to ask some questions to the authors.

(page 6, line 15) Regarding hypertension, hyperlipidemia, and diabetes mellitus, these definitions are somewhat unclear. What was the absolute value of blood pressure for the diagnosis of hypertension? Did not the authors evaluate a value of glucose for the definition of diabetes mellitus? Please clarify.

(page 6, line 21) Who determined the cause of death? Please clarify also how to provide the diagnosis. At what timing was the primary endpoint evaluated? At 3 years?

(page 8, line 6) When you determine the definition of AVCS and MVCS, did you refer to any references? If yes, please cite them here.

(page 8, line 21) The authors described “prior scientific reports”. The relevant references are required.

(page 10, Table 1) Please specify the kind of valvular diseases. Among valvular diseases with severity of ≥moderate, the impact of aortic stenosis on the primary endpoint or lower ASA would be different from that of mitral regurgitation.

Reviewers' comments:

Reviewer's Responses to Questions

**Comments to the Author**

1. Is the manuscript technically sound, and do the data support the conclusions?

Reviewer #1: Partly

Reviewer #2: Yes

2. Has the statistical analysis been performed appropriately and rigorously? 

Reviewer #1: No

Reviewer #2: Yes

3. Have the authors made all data underlying the findings in their manuscript fully available?

Reviewer #1: Yes

Reviewer #2: Yes

4. Is the manuscript presented in an intelligible fashion and written in standard English?

Reviewer #1: Yes

Reviewer #2: Yes

5. Review Comments to the Author

Reviewer #1: Comments to the Authors)

I am pleased to review this paper showing the impact of sigmoid septum on cardio-cerebrovascular outcomes in patients with chronic renal failure on hemodialysis.

Major comments)

# As original definition of the “sigmoid septum” has not always been appreciated, there have been multiple reports showing variable quantitative and qualitative measurement of the “sigmoid septum”. What authors measured here to quantify the extent of the “sigmoid septum” is the angulation between the ascending aorta and ventricular septum. This methodology is quite similar to the “aorto-ventricular angle”, which cannot correctly define the sigmoid septum, originally defined as the “evident bulge of the ventricular septum toward the left ventricular cavity”. Rather, this quantifies the angulated relationship between the ventricular septum and ascending aorta. Such angulation is independent of the extent of septal bulging, rather it just reflects the age-related (as also demonstrated by Table 3) deep aortic wedging due to aortic elongation, which cannot be measured by echocardiography. Thus, if authors utilize this measurement which did not quantify the extent of septal bulging, the title should read “Impact of Angulated Aorto-septal relationship on Cardio-cerebrovascular Outcomes in Patients Undergoing Hemodialysis”.

Ref)

Mori S, et al. J Anat. 2017;231:110-120.

Tsuda D, et al. Echocardiography. 2022;39:248-259.

# As this is an echocardiographic study, dynamic evaluation (systolic and diastolic quantification of the measurements they applied, extent of the left ventricular outflow obstruction) is recommended.

# As described above, since the event group shows higher age and higher blood pressure, it is quite reasonable to expect smaller aorto-ventricular angle. Again, such angulation is not the representative parameter of the sigmoid septum, rather it should be an indirect measurement of the potential deep aortic wedging.

Minor comments)

# Considering the size and nature of this retrospective single center study only using echocardiographic measurement, I am afraid there are too many (24!) authors. Corresponding author should double check the author contributions.

# Figure 2B and 2C should be explained in detail in the legend. Also, each panel in Figure 2C seems to show different timing (mid-diastole to end-diastole).

# “The main results did not change when each of the six clinical parameters were also included in the multivariable Cox proportional hazards analyses (Table S1).” Why, in the supplementary table, LVEF and LVMI were excluded from the analysis?

Reviewer #2: Review comments.

The manuscript is fit for publication fulfilling the journals criteria. As a minor revision a comment could be added in the discussion regarding the presence of Q waves on the ECG and their correlation with the presence of sigmoid septum. Dialysis patients are a special population with increased incidence of ventricular hypertrophy or cardiovascular diseases. So a comment could be added regarding the differentil diagnosis of the presence of Q-waves on the ECG, based on an other article entitled ‘’ Correlation between sigmoid interventricular septum angle and presence of Q waves on the electrocardiogram’’.

6. PLOS authors have the option to publish the peer review history of their article (what does this mean?). If published, this will include your full peer review and any attached files.

Reviewer #1: No

Reviewer #2: **Yes: **Afendoulis Dimitrios

---

## [Author Response · Author response to Decision Letter 0]

26 Dec 2023

Reply to the Editor

We appreciate the editor’s valuable time and comments, which have helped improve the scientific quality of the manuscript. 

Editor’s comment_1)

(page 6, line 15) Regarding hypertension, hyperlipidemia, and diabetes mellitus, these definitions are somewhat unclear. What was the absolute value of blood pressure for the diagnosis of hypertension? Did not the authors evaluate a value of glucose for the definition of diabetes mellitus? Please clarify.

Reply)

We thank the editor for pointing out the unclear definitions. As the primary endpoint was cardio-cerebrovascular disease, the definitions of risk factors were important. According to the guidelines for the management of hypertension in Japan, systemic hypertension is diagnosed when the office blood pressure is ≥ 140/90 mmHg or home blood pressure ≥ 135/85 mmHg [1]. Based on the clinical practice guidelines for diabetes in Japan, diabetes mellitus is defined as fasting blood glucose ≥126 mg/dL, greater medullary blood glucose of 200 mg/dL, or HbA1c ≥ 6.5%. Furthermore, as per the editor’s comments, the diagnosis of diabetes mellitus requires meeting at least one of the glucose criteria [2]. However, we could not assess whether the blood glucose level was fasting or medullary. Therefore, we re-reviewed all patients’ blood examinations within 1 year of routine echocardiography if the blood glucose level was ≥ 200 mg/dL to clarify the diagnosis of diabetes mellitus. However, as a result, no diagnosis of diabetes mellitus was changed. We revised the study protocol section as follows.

<Page 5, lines 17-25>

Before revised:

Hypertension, hyperlipidemia, and diabetes mellitus were included as cardiovascular risk factors. Hypertension was defined as a medical history of hypertension or being under anti-hypertensive medication; hyperlipidemia was defined as a history of hyperlipidemia, undergoing treatment for hyperlipidemia, or a serum low-density lipoprotein level ≥ 140 mg/dl at baseline; and diabetes mellitus was defined as a history of diabetes mellitus, treatment with glucose-lowering therapy, or a hemoglobin A1c level ≥ 6.5% at baseline. 

After revised:

Hypertension, hyperlipidemia, and diabetes mellitus were included as cardiovascular risk factors. Hypertension was defined as having a history of hypertension or being under anti-hypertensive treatment. A history of hypertension is diagnosed as an office blood pressure ≥ 140/90 mmHg or home blood pressure ≥ 135/85 mmHg [1]. Hyperlipidemia was defined as having a history of hyperlipidemia, taking medication for hyperlipidemia, or a serum low-density lipoprotein level ≥ 140 mg/dL at baseline. Diabetes mellitus was defined as having a history of diabetes mellitus or undergoing treatment with glucose-lowering therapy and was diagnosed by meeting the following criteria at least two times, including the blood glucose criterion at least once, the year prior to echocardiography: hemoglobin A1c ≥ 6.5% and blood glucose ≥ 200 mg/dL [2]. 

Editor’s comment_2)

(page 6, line 21) Who determined the cause of death? Please clarify also how to provide the diagnosis. At what timing was the primary endpoint evaluated? At 3 years?

Reply)

We appreciate the valuable comments regarding methodology. TN, JY, and TO carefully assessed the cardio-cerebrovascular events from medical records, including all summaries, medical information sheets, and reference letters. If details needed to be clarified, we contacted the attending hemodialysis physician at Masuko Memorial Hospital to determine an accurate endpoint. The patient’s prognosis was retrospectively verified between September and December 2021. We included events that occurred within 36 months for the time-dependent ROC. However, the other analyses used the final prognosis information obtained between September and December 2021, resulting in 56 events, including those observed after 36 months. We revised the study protocol and statistical analysis sections.

<Page 5, lines 26-28 and Page 6, lines 1-7>

Before revised:

The primary endpoint was a composite of cardio-cerebrovascular death and admission due to heart failure, myocardial ischemic event, or cerebrovascular event. A myocardial ischemic event was defined as any myocardial ischemic event that required revascularization. Cerebrovascular event was defined as cerebral infarction, cerebral hemorrhage, or subarachnoid hemorrhage. Sudden death by unexplained cause and any death associated with ventricular arrhythmia were included in cardio-cerebrovascular death. 

After revised:

The primary endpoint was a composite of admission and death due to cardio-cerebrovascular events, including heart failure, myocardial ischemic events, or cerebrovascular events. A myocardial ischemic event was defined as an admission due to ischemic heart disease that required revascularization. A cerebrovascular event was defined as cerebral infarction, cerebral hemorrhage, or subarachnoid hemorrhage. Unexplained sudden death and any death accompanied by ventricular arrhythmia were included in cardio-cerebrovascular death. Two cardiologists (TN and JY) and a nephrologist (TO) carefully assessed the cardio-cerebrovascular events from medical records, including all summaries, medical information sheets, and reference letters. If the authors needed to clarify the details, they contacted a neurologist or the attending hemodialysis physician at Masuko Memorial Hospital to determine an accurate endpoint.

<Page 8, lines 24-26>

Before revised:

None.

After revised:

As the patients’ prognoses were retrospectively verified between September and December 2021, the primary endpoint observed in the current study included the events observed after 36 months.

Editor’s comment_3)

(page 8, line 6) When you determine the definition of AVCS and MVCS, did you refer to any references? If yes, please cite them here.

Reply)

We would like to thank the editor for the question. We did not refer to any prior study to quantify AVCS and MVCS and thought it was better to discuss not only the presence or absence of calcification, but the degree to which it occurred.

Editor’s comment_4)

(page 8, line 21) The authors described “prior scientific reports”. The relevant references are required.

Reply)

We appreciate the editor’s comment and apologize for the missing information. We intended to use ASA, AVCS, and MVCS and other echocardiographic and clinical variables, including parameters considered risk factors for the primary endpoint, in the Cox regression analyses. We referred to the prior research to explain the rationale for selecting variables [3-11]. We corrected the statistical analysis section as follows.

<Page 8, lines 11-13>

Before revised:

Cox proportional hazards analyses were performed using ASA, AVCS, MVCS, and clinical variables that were considered to be risk factors for the primary endpoint by prior scientific reports.

After revised:

Cox proportional hazards analyses were performed using ASA, AVCS, MVCS, and other echocardiographic and clinical variables, including parameters that are considered risk factors for the primary endpoint [3-11]

Editor’s comment_5)

(page 10, Table 1) Please specify the kind of valvular diseases. Among valvular diseases with severity of ≥moderate, the impact of aortic stenosis on the primary endpoint or lower ASA would be different from that of mitral regurgitation.

Reply)

We appreciate the valuable comment, as this is crucial in echocardiography research. We separated valvular diseases with moderate or greater severity into each regurgitative and stenotic valvular disease. We found that mitral valve regurgitation and tricuspid valve regurgitation with moderate or greater severity occurred significantly more frequently in the event occurrence group than in the event-free group. Aortic valve stenosis of moderate or greater severity tended to occur in the event-occurrence group, but the difference between the groups was not significant, probably because of the small number of the primary endpoint. We added the information to Table 1.

 

Reply to Reviewer #1

We want to express our thanks to the reviewer for their time and valuable comments, which provided an accurate understanding of the sigmoid septum and aortic wedging and their fundamental substrates. 

Reviewer’s comment_1)

# As original definition of the “sigmoid septum” has not always been appreciated, there have been multiple reports showing variable quantitative and qualitative measurement of the “sigmoid septum”. What authors measured here to quantify the extent of the “sigmoid septum” is the angulation between the ascending aorta and ventricular septum. This methodology is quite similar to the “aorto-ventricular angle”, which cannot correctly define the sigmoid septum, originally defined as the “evident bulge of the ventricular septum toward the left ventricular cavity”. Rather, this quantifies the angulated relationship between the ventricular septum and ascending aorta. Such angulation is independent of the extent of septal bulging, rather it just reflects the age-related (as also demonstrated by Table 3) deep aortic wedging due to aortic elongation, which cannot be measured by echocardiography. Thus, if authors utilize this measurement which did not quantify the extent of septal bulging, the title should read “Impact of Angulated Aorto-septal relationship on Cardio-cerebrovascular Outcomes in Patients Undergoing Hemodialysis”.

Ref)

Mori S, et al. J Anat. 2017;231:110-120.

Tsuda D, et al. Echocardiography. 2022;39:248-259.

Reply)

We appreciate the crucially important comment and have gained a better understanding that will allow us to describe sigmoid septum and aortic wedging more accurately in future research and clinical work. We discuss the differences in substrate and clinical effect between an angulated ASA and sigmoid septum in the revised manuscript.

As per the reviewer’s comment, we investigated the impact of a narrowed aorto-septal angle on cardio-cerebrovascular events in hemodialysis patients and concluded that an angulated aorto-septal relationship impacts worse cardio-cerebrovascular outcomes. We revised the manuscript as follows.

<Page 1, lines 5-6>

Before revised:

Impact of Sigmoid Septum on Cardio-cerebrovascular Outcomes in Patients Undergoing Hemodialysis

After revised:

Impact of an Angulated Aorto-septal Relationship on Cardio-cerebrovascular Outcomes in Patients Undergoing Hemodialysis

<Page 3, lines 2-7>

Before revised:

A sigmoid septum is associated with calcified arterial and valvular changes, which are risk factors for cardio-cerebrovascular events in patients undergoing hemodialysis. However, the clinical impact of sigmoid septum in patients requiring hemodialysis has not been clarified. 

After revised:

Aortic and valvular calcification are well-known risk factors for cardio-cerebrovascular events in patients undergoing hemodialysis. We investigated the clinical impact of an angulated aorto-septal angle as a result of aortic elongation due to aortic calcification on cardio-cerebrovascular outcomes in patients undergoing hemodialysis. We investigated 306 patients (mean age 65.4 years, 68% male) who underwent pre-scheduled routine echocardiography between April and September 2018. The angle between the anterior wall of the aorta and the ventricular septal surface (ASA) was quantified.

<Page 4, lines 6-16>

Before revised:

Systemic calcification is one of the changes of aging and is also recognized to improve with the initiation of hemodialysis. Calcified valvular change and arterial calcification are also risk factors for cardio-cerebrovascular diseases in patients undergoing hemodialysis. On the other hand, ventricular sigmoid septum is considered to be associated with aortic atherosclerosis or thickening and calcification of the aortic or mitral valve. Furthemore, ventricular sigmoid septum is one of the changes with aging, with recent evidence demonstrating its association with poor cardio-cerebrovascular outcomes. Therefore, we can expect that the presence of sigmoid septum is probably a risk factor for poor cardio-cerebrovascular outcomes in patients with dialysis. However, the clinical significance has yet to be elucidated.

After revised:

Calcified valvular change and arterial calcification are well-known risk factors for cardio-cerebrovascular diseases in patients undergoing hemodialysis [12-16]. In addition, systemic calcification is one of the changes of aging and improves with the initiation of hemodialysis [17]. Aortic elongation is also a result of the aging process associated with aortic calcification [18-20]. Aortic wedging, which is considered to be accompanied by aortic elongation, could present as a narrowed angle between the left ventricular (LV) and aortic axis [21, 22]. Some recent studies demonstrated that the echocardiographic-derived angulated relationship between the ventricular septum and aorta impact poor cardio-cerebrovascular outcomes [23, 24]. Therefore, we expect that the presence of an angulated LV septum and aorta on echocardiography is a risk factor for poor cardio-cerebrovascular outcomes in patients requiring hemodialysis.

<Page 7, lines 3-7>

Before revised:

2.4 Echocardiographic quantification of the ventricular sigmoid septum

We measured the angle between the anterior wall of the aorta and the ventricular septal surface (aorto-septal angle; ASA) to assess the degree of ventricular sigmoid septum (Figure 2A).

After revised:

2.4 Echocardiographic quantification of the angle between the aorta and left ventricle

We measured the angle between the anterior wall of the aorta and the ventricular septal surface (aorto-septal angle; ASA) to assess the degree of angulation between the aorta and left ventricle (Fig 2A).

<Page 16, line 10>

Before revised:

3.3 Factors associated with sigmoid septum

After revised:

3.3 Factors associated with an angulated aorto-septal angle

<Page 20, line 9-16>

Before revised:

4.1 Sigmoid septum

Atherosclerosis, thickening of the aortic or mitral valve, and hypertension are considered to contribute to sigmoid septum. Its association with clinically adverse events and diastolic dysfunction were recently reported in non-hemodialysis patients. Generally, sigmoid septum is observed in the elderly population. In our investigations, only age was an independent determinant of sigmoid septum defined as ASA ≤119.4 degrees (Table 3). However, as both the presence of sigmoid septum and higher age were the independent risk factors for the cardio-cerebrovascular events, sigmoid septum must have other unexplained factors causing adverse events.

After revised:

4.1 Angulated aorto-septal angle

The clinical evidence for an angulated ASA is still small but has recently grown. An association with clinically adverse events and diastolic dysfunction has been reported in non-hemodialysis patients [23-25]. Aortic wedging that correlates with an angulated LV-aorta angle is strongly associated with age [22]. Our investigations also showed that only age is an independent determinant of ASA ≤ 119.4 degrees (Table 3). However, as both the presence of an angulated ASA and higher age were independent risk factors for cardio-cerebrovascular events, an angulated ASA must have unexplained factors other than age causing adverse events.

<Page 20, line 20-22>

Before revised:

None.

After revised:

As AVCS and MVCS were significantly associated with a narrowed ASA in the univariable logistic regression analysis, valvular calcification may morphologically contribute to the angulated ASA during the aging process.

<Page 20, line 25-27>

Before revised:

Thus, the presence of sigmoid septum must be a surrogate marker of negative factors that accumulate and are affected not only by aging, but also dialysis-associated changes in calcification and tissue injury. 

After revised:

Thus, the presence of an angulated ASA must be a surrogate marker of negative factors that accumulate and are affected not only by aging, but also dialysis-associated changes in calcification and tissue injury. 

<Page 20, lines 28-29 and Page 21, line 1-12>

Before revised:

None.

After revised:

On the other hand, sigmoid septum is a well-known morphological change similar to, but different from, an angulated ASA. Initially, sigmoid septum was defined as the base of the ventricular septum protruding toward the LV cavity [26]. Sometimes variously termed subaortic ventricular septal bulge and discrete upper septal hypertrophy, sigmoid septum is associated with age [27-29], with atherosclerosis, thickening of the aortic or mitral valve, and hypertension considered to contribute to its presence [30-32]. Otherwise, sigmoid septum has no impact on the cardiovascular and mortality risk in patients with heart failure and on exercise tolerance in healthy populations [33, 34]. Interestingly, an angulated relationship between the ventricular septum and ascending aorta did not correlate with the extent of septal bulge in prior research using computed tomography [22], suggesting that the underlying substrates and pathological roles are different between an angulated ASA and sigmoid septum. For example, aortic elongation is not always accompanied by atherosclerosis [19,20]. Referring to the evidence above and the results of the current study, an angulated ASA is likely to affect the clinical prognosis compared with sigmoid septum. Further research investigating the difference in contributing factors and influence between an angulated ASA and sigmoid septum in various groups of patients is of interest.

<Page 23, lines 5-7>

Before revised:

The presence of echocardiography-proven sigmoid septum and greater area of calcification on the aortic valve were independent determinants of cardio-cerebrovascular events in patients requiring hemodialysis.

After revised:

The presence of an echocardiography-derived angulated ASA and greater area of calcification on the aortic valve were independent determinants of cardio-cerebrovascular events in patients requiring hemodialysis.

Reviewer’s comment_2)

# As this is an echocardiographic study, dynamic evaluation (systolic and diastolic quantification of the measurements they applied, extent of the left ventricular outflow obstruction) is recommended.

Reply)

In the current study, no patient had a left ventricular outlet tract (LVOT) obstruction with systolic anterior-mitral valve leaflet motion and pressure gradient ≥ 50 mmHg [35] at rest. However, the LVOT pressure gradient is not routinely quantified in routine echocardiography at Masuko Memorial Hospital. Therefore, we cannot assess the LVOT obstruction as a continuous value in this retrospective study. As the reviewer commented, because the current research investigated the left ventricular morphology, the LVOT pressure gradient is the parameter needed to discuss the mechanisms contributing to the worse outcomes. We added the information to the limitation section.

<Page 20, lines 22-25>

Before revised:

In the current study, no patients had a significant LV outlet obstruction with systolic anterior-mitral valve leaflet motion during routine echocardiographic examination, and LV outlet obstruction was not associated with our results.

After revised:

In the current study, no patients had a significant LV outlet tract obstruction (pressure gradient ≥ 50 mmHg [35]) with systolic anterior-mitral valve leaflet motion during routine echocardiographic examination, and LV outlet tract obstruction was not associated with our results.

<Page 22, lines 26-29 and page 23, lines 1>

Before revised:

None.

After revised:

As the LVOT pressure gradient was quantified in only necessary patients, we could not perform a dynamic evaluation of the extent of LVOT narrowing. Furthermore, because exercise or dobutamine stress echocardiography were not performed at Mesuko Memorial Hospital, the relationship between the ASA and subclinical LVOT obstruction was not assessed.

Reviewer’s comment_3)

# As described above, since the event group shows higher age and higher blood pressure, it is quite reasonable to expect smaller aorto-ventricular angle. Again, such angulation is not the representative parameter of the sigmoid septum, rather it should be an indirect measurement of the potential deep aortic wedging.

Reply)

We thank the reviewer for the kind comments repeated in the review process. We have tried to reflect our understanding of aortic wedging in the manuscript.

Minor comments)

Reviewer’s comment_4)

# Considering the size and nature of this retrospective single center study only using echocardiographic measurement, I am afraid there are too many (24!) authors. Corresponding author should double check the author contributions.

Reply)

We understand what the reviewer is pointing out by mentioning the large number of co-authors. The first author’s group is conducting serial and multifaceted research investigating patients with hemodialysis. Two manuscripts were accepted, and now another manuscript using another dataset is under review by another journal. The dataset includes various parameters and was obtained, double-checked, discussed, and reviewed critically by all authors. The corresponding author has verified the contributions of all co-authors to the current study and believes the work of all co-authors was necessary. The corresponding author would greatly appreciate it if the reviewer allows the nomination of all co-authors.

Reviewer’s comment_5)

# Figure 2B and 2C should be explained in detail in the legend. Also, each panel in Figure 2C seems to show different timing (mid-diastole to end-diastole).

Reply)

We thank the reviewer for this recommendation, as briefly sharing the process of quantifying the score with the reader would increase understanding. Per the reviewer’s advice, we added an explanation of how to score AVCS and MVCS in the figure legends. 

Regarding the different timing between panels in Figure 2C, we unified the panels to mid-diastolic phase, which is the most common phase for opening of the mitral valve.

Reviewer’s comment_6)

# “The main results did not change when each of the six clinical parameters were also included in the multivariable Cox proportional hazards analyses (Table S1).” Why, in the supplementary table, LVEF and LVMI were excluded from the analysis?

Reply)

We appreciate the question. LVEF and LVMI were included in the additional Cox proportional hazards analyses, but we omitted the results for LVEF and LVMI from the table to avoid redundant presentation. Based on the reviewer’s comment, we found it better not to omit these data and present them in the supplementary table.

 

Reply to Reviewer #2

We appreciate the reviewer’s kind rating. The comment encourages us to attempt further research. 

Reviewer’s comment_1)

The manuscript is fit for publication fulfilling the journals criteria. As a minor revision a comment could be added in the discussion regarding the presence of Q waves on the ECG and their correlation with the presence of sigmoid septum. Dialysis patients are a special population with increased incidence of ventricular hypertrophy or cardiovascular diseases. So a comment could be added regarding the differentil diagnosis of the presence of Q-waves on the ECG, based on an other article entitled ‘’ Correlation between sigmoid interventricular septum angle and presence of Q waves on the electrocardiogram’’.

Reply)

Before seeing the reviewer’s comment, we had no idea about the relationship between electrocardiography findings and sigmoid septum and appreciate bringing up a new concept.

Electrocardiography is routinely examined every 3 or 4 months at Masuko Memorial Hospital. We assessed the presence or absence of Q wave in V1 and V2 leads in all patients using the last electrocardiography before echocardiography. As a result, the presence of Q wave in both V1 and V2 tended to be associated with a smaller ASA, but the relationship was not significant in the current patient cohort. The reasons for the above results are possibly explained by the small number of participants; ASA was not a representative measurement of sigmoid septum. We added a discussion of this topic in the Discussion section.

<Page 21 lines 13-22>

Before revised:

None.

After revised:

Sigmoid septum is associated with the presence of Q waves in V1 and V2 leads [36]. As findings of ischemic heart disease are clinically essential for patients undergoing dialysis because of their high prevalence of ischemic heart disease [37, 38], we investigated the association between ASA and the presence of Q wave in V1-2. As a result, the average ASAs were 114.0 ± 13.1 degrees and 118.2 ± 11.3 degrees in patients with Q wave in both V1-2 leads (n = 20) and patients without Q wave (n = 286), and the difference was not significant (P = 0.12). The above results may be affected by the small number of participants; ASA was not a representative measurement of sigmoid septum. The presence of a non-ischemic Q wave in dialysis patients is essential to clinically differentiate in this cohort; thus, further studies in a larger group of patients undergoing dialysis are needed to investigate the relationship between the representative measurements of sigmoid septum and Q wave in V1-2.

 

References

1. Umemura S, Arima H, Arima S, Asayama K, Dohi Y, Hirooka Y, et al. The Japanese Society of Hypertension Guidelines for the Management of Hypertension (JSH 2019). Hypertens Res. 2019;42(9):1235-481. Epub 2019/08/04. doi: 10.1038/s41440-019-0284-9. PubMed PMID: 31375757.

2. Araki E, Goto A, Kondo T, Noda M, Noto H, Origasa H, et al. Japanese Clinical Practice Guideline for Diabetes 2019. J Diabetes Investig. 2020;11(4):1020-76. Epub 2020/10/07. doi: 10.1111/jdi.13306. PubMed PMID: 33021749; PubMed Central PMCID: PMCPMC7378414.

3. Nakayama T, Yamamoto J, Ozeki T, Yasuda K, Yamazaki C, Ito T, et al. Impact of left ventricular hypertrophy on clinical outcomes in patients with dialysis: a single-center study in Japan. J Med Ultrason (2001). 2022. Epub 2022/03/18. doi: 10.1007/s10396-022-01197-4. PubMed PMID: 35298744.

4. Inaguma D, Koide S, Takahashi K, Hayashi H, Hasegawa M, Yuzawa Y. Association between resting heart rate just before starting the first dialysis session and mortality: A multicentre prospective cohort study. Nephrology (Carlton). 2018;23(5):461-8. Epub 2017/03/24. doi: 10.1111/nep.13048. PubMed PMID: 28332737.

5. Parfrey PS, Foley RN, Harnett JD, Kent GM, Murray DC, Barre PE. Outcome and risk factors for left ventricular disorders in chronic uraemia. Nephrol Dial Transplant. 1996;11(7):1277-85. Epub 1996/07/01. PubMed PMID: 8672023.

6. Payne J, Sharma S, De Leon D, Lu JL, Alemu F, Balogun RA, et al. Association of echocardiographic abnormalities with mortality in men with non-dialysis-dependent chronic kidney disease. Nephrol Dial Transplant. 2012;27(2):694-700. Epub 2011/05/27. doi: 10.1093/ndt/gfr282. PubMed PMID: 21613387; PubMed Central PMCID: PMCPMC3350343.

7. Kopple JD. Nutritional status as a predictor of morbidity and mortality in maintenance dialysis patients. Asaio j. 1997;43(3):246-50. Epub 1997/05/01. PubMed PMID: 9152503.

8. Fouque D, Kalantar-Zadeh K, Kopple J, Cano N, Chauveau P, Cuppari L, et al. A proposed nomenclature and diagnostic criteria for protein-energy wasting in acute and chronic kidney disease. Kidney Int. 2008;73(4):391-8. Epub 2007/12/21. doi: 10.1038/sj.ki.5002585. PubMed PMID: 18094682.

9. Waikar SS, Curhan GC, Brunelli SM. Mortality associated with low serum sodium concentration in maintenance hemodialysis. Am J Med. 2011;124(1):77-84. Epub 2010/12/29. doi: 10.1016/j.amjmed.2010.07.029. PubMed PMID: 21187188; PubMed Central PMCID: PMCPMC3040578.

10. Rhee CM, Ravel VA, Ayus JC, Sim JJ, Streja E, Mehrotra R, et al. Pre-dialysis serum sodium and mortality in a national incident hemodialysis cohort. Nephrol Dial Transplant. 2016;31(6):992-1001. Epub 2015/09/28. doi: 10.1093/ndt/gfv341. PubMed PMID: 26410882; PubMed Central PMCID: PMCPMC4876967.

11. Dekker MJ, Marcelli D, Canaud B, Konings CJ, Leunissen KM, Levin NW, et al. Unraveling the relationship between mortality, hyponatremia, inflammation and malnutrition in hemodialysis patients: results from the international MONDO initiative. Eur J Clin Nutr. 2016;70(7):779-84. Epub 2016/04/21. doi: 10.1038/ejcn.2016.49. PubMed PMID: 27094625.

12. Wang AY, Wang M, Woo J, Lam CW, Li PK, Lui SF, et al. Cardiac valve calcification as an important predictor for all-cause mortality and cardiovascular mortality in long-term peritoneal dialysis patients: a prospective study. J Am Soc Nephrol. 2003;14(1):159-68. Epub 2002/12/31. doi: 10.1097/01.asn.0000038685.95946.83. PubMed PMID: 12506148.

13. Raggi P, Bellasi A, Gamboa C, Ferramosca E, Ratti C, Block GA, et al. All-cause mortality in hemodialysis patients with heart valve calcification. Clin J Am Soc Nephrol. 2011;6(8):1990-5. Epub 2011/06/28. doi: 10.2215/cjn.01140211. PubMed PMID: 21700824; PubMed Central PMCID: PMCPMC3359535.

14. Wang Z, Jiang A, Wei F, Chen H. Cardiac valve calcification and risk of cardiovascular or all-cause mortality in dialysis patients: a meta-analysis. BMC Cardiovasc Disord. 2018;18(1):12. Epub 2018/01/27. doi: 10.1186/s12872-018-0747-y. PubMed PMID: 29370754; PubMed Central PMCID: PMCPMC5785897.

15. London GM, Guérin AP, Marchais SJ, Métivier F, Pannier B, Adda H. Arterial media calcification in end-stage renal disease: impact on all-cause and cardiovascular mortality. Nephrol Dial Transplant. 2003;18(9):1731-40. Epub 2003/08/26. doi: 10.1093/ndt/gfg414. PubMed PMID: 12937218.

16. Zhang A, Wang S, Li H, Yang J, Wu H. Aortic arch calcification and risk of cardiovascular or all-cause and mortality in dialysis patients: A meta-analysis. Sci Rep. 2016;6:35375. Epub 2016/10/18. doi: 10.1038/srep35375. PubMed PMID: 27748417; PubMed Central PMCID: PMCPMC5066315.

17. Ponte B, Pruijm M, Pasch A, Dufey-Teso A, Martin PY, de Seigneux S. Dialysis initiation improves calcification propensity. Nephrol Dial Transplant. 2020;35(3):495-502. Epub 2019/11/19. doi: 10.1093/ndt/gfz222. PubMed PMID: 31738424.

18. Adriaans BP, Heuts S, Gerretsen S, Cheriex EC, Vos R, Natour E, et al. Aortic elongation part I: the normal aortic ageing process. Heart. 2018;104(21):1772-7. Epub 2018/03/30. doi: 10.1136/heartjnl-2017-312866. PubMed PMID: 29593078.

19. Ferrari AU, Radaelli A, Centola M. Invited review: aging and the cardiovascular system. J Appl Physiol (1985). 2003;95(6):2591-7. Epub 2003/11/06. doi: 10.1152/japplphysiol.00601.2003. PubMed PMID: 14600164.

20. Sugawara J, Hayashi K, Yokoi T, Tanaka H. Age-associated elongation of the ascending aorta in adults. JACC Cardiovasc Imaging. 2008;1(6):739-48. Epub 2009/04/10. doi: 10.1016/j.jcmg.2008.06.010. PubMed PMID: 19356510.

21. Mori S, Yamashita T, Takaya T, Kinugasa M, Takamine S, Shigeru M, et al. Association between the rotation and three-dimensional tortuosity of the proximal ascending aorta. Clin Anat. 2014;27(8):1200-11. Epub 2014/08/06. doi: 10.1002/ca.22452. PubMed PMID: 25091125.

22. Mori S, Anderson RH, Takaya T, Toba T, Ito T, Fujiwara S, et al. The association between wedging of the aorta and cardiac structural anatomy as revealed using multidetector-row computed tomography. J Anat. 2017;231(1):110-20. Epub 2017/04/12. doi: 10.1111/joa.12611. PubMed PMID: 28397961; PubMed Central PMCID: PMCPMC5472522.

23. Nakayama T, Oshima Y, Shintani Y, Yamamoto J, Yokoi M, Ito T, et al. Ventricular Sigmoid Septum as a Risk Factor for Anthracycline-Induced Cancer Therapeutics-Related Cardiac Dysfunction in Patients With Malignant Lymphoma. Circ Rep. 2022;4(4):173-82. Epub 2022/04/19. doi: 10.1253/circrep.CR-21-0145. PubMed PMID: 35434414; PubMed Central PMCID: PMCPMC8977195.

24. Manea P, Ghiuru R. Correlations between the presence of sigmoid interventricular septum and increased relapse risk of stroke in hypertensive patients. Rev Med Chir Soc Med Nat Iasi. 2013;117(4):857-62. Epub 2014/02/08. PubMed PMID: 24502061.

25. Okada K, Mikami T, Kaga S, Nakabachi M, Abe A, Yokoyama S, et al. Decreased aorto-septal angle may contribute to left ventricular diastolic dysfunction in healthy subjects. J Clin Ultrasound. 2014;42(6):341-7. Epub 2014/01/18. doi: 10.1002/jcu.22126. PubMed PMID: 24436178.

26. Goor D, Lillehei CW, Edwards JE. The "sigmoid septum". Variation in the contour of the left ventricular outt. Am J Roentgenol Radium Ther Nucl Med. 1969;107(2):366-76. Epub 1969/10/01. PubMed PMID: 4241898.

27. Krasnow N. Subaortic septal bulge simulates hypertrophic cardiomyopathy by angulation of the septum with age, independent of focal hypertrophy. An echocardiographic study. J Am Soc Echocardiogr. 1997;10(5):545-55. Epub 1997/06/01. doi: 10.1016/s0894-7317(97)70009-9. PubMed PMID: 9203495.

28. Swinne CJ, Shapiro EP, Jamart J, Fleg JL. Age-associated changes in left ventricular outflow tract geometry in normal subjects. Am J Cardiol. 1996;78(9):1070-3. Epub 1996/11/01. doi: 10.1016/s0002-9149(96)00542-5. PubMed PMID: 8916496.

29. Toth AB, Engel JA, McManus AM, McManus BM. Sigmoidity of the ventricular septum revisited: progression in early adulthood, predominance in men, and independence from cardiac mass. Am J Cardiovasc Pathol. 1988;2(3):211-23. Epub 1988/01/01. PubMed PMID: 3219204.

30. Ieki K, Imataka K, Sakurai S, Okamoto E, Ashida T, Fujii J. [Differentiation of hypertrophic cardiomyopathy and hypertensive cardiac hypertrophy using the patterns of interventricular septum hypertrophy]. J Cardiol. 1996;27(6):309-14. Epub 1996/06/01. PubMed PMID: 9062591.

31. Chen-Tournoux A, Fifer MA, Picard MH, Hung J. Use of tissue Doppler to distinguish discrete upper ventricular septal hypertrophy from obstructive hypertrophic cardiomyopathy. Am J Cardiol. 2008;101(10):1498-503. Epub 2008/05/13. doi: 10.1016/j.amjcard.2008.01.027. PubMed PMID: 18471465.

32. Belenkie I, MacDonald RP, Smith ER. Localized septal hypertrophy: part of the spectrum of hypertrophic cardiomyopathy or an incidental echocardiographic finding? Am Heart J. 1988;115(2):385-90. Epub 1988/02/01. doi: 10.1016/0002-8703(88)90486-3. PubMed PMID: 3341173.

33. Diaz T, Pencina MJ, Benjamin EJ, Aragam J, Fuller DL, Pencina KM, et al. Prevalence, clinical correlates, and prognosis of discrete upper septal thickening on echocardiography: the Framingham Heart Study. Echocardiography. 2009;26(3):247-53. Epub 2009/01/30. doi: 10.1111/j.1540-8175.2008.00806.x. PubMed PMID: 19175779; PubMed Central PMCID: PMCPMC2657181.

34. Canepa M, Malti O, David M, AlGhatrif M, Strait JB, Ameri P, et al. Prevalence, clinical correlates, and functional impact of subaortic ventricular septal bulge (from the Baltimore Longitudinal Study of Aging). Am J Cardiol. 2014;114(5):796-802. Epub 2014/08/19. doi: 10.1016/j.amjcard.2014.05.068. PubMed PMID: 25129067; PubMed Central PMCID: PMCPMC4495884.

35. Kitaoka H, Tsutsui H, Kubo T, Ide T, Chikamori T, Fukuda K, et al. JCS/JHFS 2018 Guideline on the Diagnosis and Treatment of Cardiomyopathies. Circ J. 2021;85(9):1590-689. Epub 2021/07/27. doi: 10.1253/circj.CJ-20-0910. PubMed PMID: 34305070.

36. Kartalis A, Afendoulis D, Moutafi M, Papagianni N, Ampeliotis M, Garoufalis S, et al. Correlation between sigmoid interventricular septum angle and presence of Q waves on the electrocardiogram. Kardiol Pol. 2022;80(9):940-2. Epub 2022/07/28. doi: 10.33963/KP.a2022.0175. PubMed PMID: 35892249.

37. Ohtake T, Kobayashi S, Moriya H, Negishi K, Okamoto K, Maesato K, et al. High prevalence of occult coronary artery stenosis in patients with chronic kidney disease at the initiation of renal replacement therapy: an angiographic examination. J Am Soc Nephrol. 2005;16(4):1141-8. Epub 2005/03/04. doi: 10.1681/asn.2004090765. PubMed PMID: 15743997.

38. Joki N, Hase H, Nakamura R, Yamaguchi T. Onset of coronary artery disease prior to initiation of haemodialysis in patients with end-stage renal disease. Nephrol Dial Transplant. 1997;12(4):718-23. Epub 1997/04/01. doi: 10.1093/ndt/12.4.718. PubMed PMID: 9141000.

---

## [Decision Letter · Decision Letter 1]

12 Jan 2024

PONE-D-23-33216R1Impact of an Angulated Aorto-septal Relationship on Cardio-cerebrovascular Outcomes in Patients Undergoing HemodialysisPLOS ONE

Dear Dr. Nakayama,

Thank you for submitting your manuscript to PLOS ONE. After careful consideration, we feel that it has merit but does not fully meet PLOS ONE’s publication criteria as it currently stands. Therefore, we invite you to submit a revised version of the manuscript that addresses the points raised during the review process.

We look forward to receiving your revised manuscript.

Kind regards,

Satoshi Higuchi

Academic Editor

PLOS ONE

Journal Requirements:

Additional Editor Comments:

Thank you very much for your effort and polite response. As the reviewer 1 noted, the current manuscript has been improved. However, there is a comment from the Editor.

According to Figure 3, censoring appears to have occurred in many cases after 36 months. It may be inappropriate to assess endpoints beyond this 36-month period.

Reviewers' comments:

Reviewer's Responses to Questions

**Comments to the Author**

1. If the authors have adequately addressed your comments raised in a previous round of review and you feel that this manuscript is now acceptable for publication, you may indicate that here to bypass the “Comments to the Author” section, enter your conflict of interest statement in the “Confidential to Editor” section, and submit your "Accept" recommendation.

Reviewer #1: (No Response)

2. Is the manuscript technically sound, and do the data support the conclusions?

Reviewer #1: Yes

3. Has the statistical analysis been performed appropriately and rigorously? 

Reviewer #1: Yes

4. Have the authors made all data underlying the findings in their manuscript fully available?

Reviewer #1: Yes

5. Is the manuscript presented in an intelligible fashion and written in standard English?

Reviewer #1: No

6. Review Comments to the Author

Reviewer #1: Comments to the Authors)

I am pleased to re-review this paper showing the impact of angulated aorto-septal relationship on cardio-cerebrovascular outcomes in patients with chronic renal failure on hemodialysis. The manuscript has been extensively revised and significantly improved, and I have only a few minor comments.

Minor comments)

# It seems that the author’s group had published multiple papers focusing on the “sigmoid septum”. If their previous methodologies used to evaluate the “sigmoid septum” were similar to the one utilized in this study, it would be better to explain that “angulated aorto-septal relationship” they used in this study is the same phenotypes that they referred to as the “sigmoid septum” in their previous studies, and the reason they changed the terminology. I believe that the authors now clearly appreciate the difference between these two concepts.

# “Sometimes variously termed subaortic ventricular septal bulge and discrete upper septal hypertrophy, sigmoid septum is associated with age [27-29], with atherosclerosis, thickening of the aortic or mitral valve, and hypertension considered to contribute to its presence [30-32].” This sentence does not make sense, I mean overall English of this paper needs professional English editing service before resubmission.

# “Initially, sigmoid septum was defined as the base of the ventricular septum protruding toward the LV cavity [26].” I think the original study that defined the “sigmoid septum” was the one by Goor D, et al. (Am J Roentgenol Radium Ther Nucl Med. 1969;107:366–376.)

# “Interestingly, an angulated relationship between the ventricular septum and ascending aorta did not correlate with the extent of septal bulge in prior research using computed tomography [22],” I am afraid citation of this sentence might be incorrect. This sentence seems to represent the findings by Tsuda et al (Echocardiography. 2022;39:248-259.).

7. PLOS authors have the option to publish the peer review history of their article (what does this mean?). If published, this will include your full peer review and any attached files.

Reviewer #1: No

---

## [Author Response · Author response to Decision Letter 1]

24 Jan 2024

Reply to the Editor

We want to thank the editor for the warm response. With the comments, we were able to update our knowledge for the research analysis.

Editor’s comment_1)

According to Figure 3, censoring appears to have occurred in many cases after 36 months. It may be inappropriate to assess endpoints beyond this 36-month period.

Reply)

We re-assessed the endpoints within 36 months. The number for the primary endpoint decreased from 56 to 54. We updated the baseline characteristics (Table 1), Cox proportional hazard analyses (Table 2, Table S1A-S1I, Table S2B, and S3B), and Kaplan-Meier analysis using the new endpoint information, and we verified that there was no need to change the conclusions. Before the correction, the Kaplan-Meier curve was drawn as 30 days per month. After the correction, the Kaplan-Meier curve was 365 days per year (182 days per 6 months), resulting in a slight change in the number at risk. 

Moreover, at the individual's request, the sentence "H.H. reviewed and supervised the statistical analyses of the current study" has been removed. H.H. strictly supervised our statistical analyses in the original and revised version of the manuscript, but his policy is not to make this statement in the manuscript.

<Page 3, lines 11-19>

Before revised:

The primary endpoint was observed in 56 patients during the observational period (median 1183 days). Multivariable Cox proportional hazards analyses identified left ventricular ejection fraction (per 10% increase: hazard ratio [HR] 0.68; 95% confidential interval [CI] 0.54-0.85, P = 0.001), left ventricular mass index (per 10 g/m2 increase: HR 1.13; 95% CI 1.05-1.23, P = 0.002), ASA (per 10 degree increase: HR 0.71; 95% CI 0.56-0.91; P = 0.006), and aortic valve calcification score (HR 1.16; 95% CI 1.06-1.28, P = 0.002) as independent determinants of the primary endpoint. Kaplan-Meier analysis showed a higher incidence of the primary endpoint in patients with ASA <119.4 degrees than those with ASA ≥119.4 degrees (Log-rank P = 0.001).

After revised:

The primary endpoint was observed in 54 patients during the observational period (median 1095 days). Multivariable Cox proportional hazards analyses identified left ventricular ejection fraction (per 10% increase: hazard ratio [HR] 0.67; 95% confidential interval [CI] 0.53-0.84, P = 0.001), left ventricular mass index (per 10 g/m2 increase: HR 1.14; 95% CI 1.05-1.24, P = 0.001), ASA (per 10 degree increase: HR 0.69; 95% CI 0.54-0.88; P = 0.003), and aortic valve calcification score (HR 1.15; 95% CI 1.04-1.26, P = 0.005) as independent determinants of the primary endpoint. Kaplan-Meier analysis showed a higher incidence of the primary endpoint in patients with ASA <119.4 degrees than those with ASA ≥119.4 degrees (Log-rank P < 0.001).

<Page 8, lines 24-25>

Before revised:

As the patients’ prognoses were retrospectively verified between September and December 2021, the primary endpoint observed in the current study included the events observed after 36 months.

After revised:

In this retrospective study, patient prognosis was evaluated up to 36 months.

<Page 12, lines 3-4>

Before revised:

The primary endpoint was observed in 56 patients during the observational period, which was a median 1183 days (interquartile range, 908-1196 days).

After revised:

The primary endpoint was observed in 54 patients during the observational period, which was a median 1095 days (interquartile range, 908-1095 days).

<Page 12, lines 7-12>

Before revised:

Multivariable Cox proportional hazards analyses identified LVEF (per 10% increase: hazard ratio [HR] 0.68; 95% confidential interval [CI] 0.54-0.85; P = 0.001), LV mass index (per 10 g/m2 increase: HR 1.13; 95% CI 1.05-1.23; P = 0.002), ASA (per 10 degree increase; HR 0.71; 95% CI 0.56-0.91; P = 0.006), and AVCS (HR 1.16; 95% CI 1.06-1.28; P = 0.002) as independent determinants of the primary endpoint (Table 2).

After revised:

Multivariable Cox proportional hazards analyses identified LVEF (per 10% increase: hazard ratio [HR] 0.67; 95% confidential interval [CI] 0.53-0.84; P = 0.001), LV mass index (per 10 g/m2 increase: HR 1.14; 95% CI 1.05-1.24; P = 0.001), ASA (per 10 degree increase; HR 0.69; 95% CI 0.54-0.88; P = 0.003), and AVCS (HR 1.15; 95% CI 1.04-1.26; P = 0.005) as independent determinants of the primary endpoint (Table 2).

<Page 9, lines 2>

Before revised:

H.H. reviewed and supervised the statistical analyses of the current study.

After revised:

Deleted.

 

Reply to the Reviewer #1

We appreciate the reviewer's careful, kind, and sincere review. The constructive comments and scientific points were crucial for our future work.

Reviewer’s comment_1)

Minor comments)

# It seems that the author’s group had published multiple papers focusing on the “sigmoid septum”. If their previous methodologies used to evaluate the “sigmoid septum” were similar to the one utilized in this study, it would be better to explain that “angulated aorto-septal relationship” they used in this study is the same phenotypes that they referred to as the “sigmoid septum” in their previous studies, and the reason they changed the terminology. I believe that the authors now clearly appreciate the difference between these two concepts.

Reply)

We thank the reviewer for the scientifically important comment. We must mention and explain why AVA was used as the representative of aortic wedging due to aortic elongation, which was different from the representative of sigmoid septum used in our previous investigations. The reason was our updated understanding of the difference between sigmoid septum and aortic wedging (because of the reviewer’s valuable comments).

We added sentences explaining this in the Discussion section.

<Page 21, lines 12-17>

Before revised:

None.

After revised:

In a previous study, we used ASA as a representative parameter of sigmoid septum. However, as we updated our knowledge with the latest research, we came to appreciate these two different morphological concepts; it is correct to use ASA to indicate aortic wedging due to aortic elongation. Aortic wedging and sigmoid septum look similar but, as mentioned above, these two concepts must be distinguished regarding different substrates and likely different clinical impacts.

Reviewer’s comment_2)

# “Sometimes variously termed subaortic ventricular septal bulge and discrete upper septal hypertrophy, sigmoid septum is associated with age [27-29], with atherosclerosis, thickening of the aortic or mitral valve, and hypertension considered to contribute to its presence [30-32].” This sentence does not make sense, I mean overall English of this paper needs professional English editing service before resubmission.

Reply)

We apologize for our insufficient English. Our manuscript underwent editing by a professional native English speaker twice, before the first submission and before submission of the revision. However, as the reviewer was concerned, we changed some sentences after the English editing. The changed sentences were edited by a professional English editor prior to this resubmission.

<Page 21, lines 1-4>

Before revised:

Sometimes variously termed subaortic ventricular septal bulge and discrete upper septal hypertrophy, sigmoid septum is associated with age, with atherosclerosis, thickening of the aortic or mitral valve, and hypertension considered to contribute to its presence.

After revised:

Sigmoid septum, which is sometimes referred to as subaortic ventricular septal bulge or discrete upper septal hypertrophy, is associated with age. In addition, atherosclerosis, thickening of the aortic or mitral valve, and hypertension are considered to contribute to the presence of sigmoid septum.

Reviewer’s comment_3)

# “Initially, sigmoid septum was defined as the base of the ventricular septum protruding toward the LV cavity [26].” I think the original study that defined the “sigmoid septum” was the one by Goor D, et al. (Am J Roentgenol Radium Ther Nucl Med. 1969;107:366–376.)

Reply)

We appreciate the reviewer pointing this out and apologize for the confusing description. We used reference number [26] to cite the sigmoid septum in the "Response to Reviewer" file according to the appearance order in the "Response to Reviewer," and we used reference number [39] to cite the same reference in the manuscript. This reference was the same as the one provided by the reviewer. 

Reviewer’s comment_4)

# “Interestingly, an angulated relationship between the ventricular septum and ascending aorta did not correlate with the extent of septal bulge in prior research using computed tomography [22],” I am afraid citation of this sentence might be incorrect. This sentence seems to represent the findings by Tsuda et al (Echocardiography. 2022;39:248-259.).

Reply)

We want to thank the reviewer for providing an accurate citation. We verified that the findings by Tsuda et al. are more suitable than ours (Mori et al. J Anat. 2017;231(1):110-20.) and replaced the reference.

---

## [Decision Letter · Decision Letter 2]

29 Jan 2024

Impact of an Angulated Aorto-septal Relationship on Cardio-cerebrovascular Outcomes in Patients Undergoing Hemodialysis

PONE-D-23-33216R2

Dear Dr. Nakayama,

We’re pleased to inform you that your manuscript has been judged scientifically suitable for publication and will be formally accepted for publication once it meets all outstanding technical requirements.

Kind regards,

Satoshi Higuchi

Academic Editor

PLOS ONE

Additional Editor Comments: Thank you very much for your polite response. Your effort and patience have notably enhanced the quality of the manuscript. Thank you again for your contribution.

Reviewers' comments:

Reviewer's Responses to Questions

**Comments to the Author**

1. If the authors have adequately addressed your comments raised in a previous round of review and you feel that this manuscript is now acceptable for publication, you may indicate that here to bypass the “Comments to the Author” section, enter your conflict of interest statement in the “Confidential to Editor” section, and submit your "Accept" recommendation.

Reviewer #1: All comments have been addressed

2. Is the manuscript technically sound, and do the data support the conclusions?

Reviewer #1: Yes

3. Has the statistical analysis been performed appropriately and rigorously? 

Reviewer #1: Yes

4. Have the authors made all data underlying the findings in their manuscript fully available?

Reviewer #1: Yes

5. Is the manuscript presented in an intelligible fashion and written in standard English?

Reviewer #1: Yes

6. Review Comments to the Author

Reviewer #1: I would like to commend extensive effort by the authors during these multiple, potentially painful revision processes. Now it looks far better than the initial one, and I have no more comments. Congratulations!

7. PLOS authors have the option to publish the peer review history of their article (what does this mean?). If published, this will include your full peer review and any attached files.

Reviewer #1: No

---

## [Editor Report · Acceptance letter]

15 Feb 2024

PONE-D-23-33216R2 

PLOS ONE

Dear Dr. Nakayama, 

I'm pleased to inform you that your manuscript has been deemed suitable for publication in PLOS ONE. Congratulations! Your manuscript is now being handed over to our production team.

Kind regards, 

on behalf of

Dr. Satoshi Higuchi 

Academic Editor

PLOS ONE